# Butyrate extends health and lifespan in mice with mitochondrial deficiency

Mitochondrial diseases progressively lead to multisystemic failure with treatment options remaining extremely limited. Here, to investigate strategies that alleviate mitochondrial dysfunction, we first generate a ubiquitous and tamoxifen-inducible knockout mouse model of mitochondrial transcription factor A (TFAM), a nuclear-encoded protein involved in mitochondrial DNA (mtDNA) maintenance − $Tfam^{fl/fl}Ubc^{Cre-ERT2}$ (iTfamKO) mice. Systemic TFAM deficiency triggers mitochondrial decline in a myriad of tissues in adult mice. Consequently, iTfamKO mice manifest multiorgan dysfunction including lipodystrophy, sarcopenia, metabolic alterations, kidney failure, neurodegeneration, and locomotor dysregulation, which result in the premature death of these mice. Interestingly, iTfamKO mice display intestinal barrier disruption and gut dysbiosis, with diminished levels of microbiota-derived short-chain fatty acids (SCFAs), such as butyrate. Mice with a deficient proof-reading version of the mtDNA polymerase gamma (mtDNA-mutator mice) phenocopy the dysfunction of the intestinal barrier and bacterial dysbiosis with reduced levels of butyrate, suggesting that different mouse models of mitochondrial dysfunction share insufficient generation of butyrate. Transfer of microbiota from healthy control mice or administration of tributyrin, a butyrate precursor, delay multiple signs of multimorbidity, extending lifespan in iTfamKO mice. Mechanistically, butyrate supplementation recovers epigenetic histone acylation marks that are lost in the intestine of *Tfam* deficient mice. Overall, our findings highlight the relevance of preserving host-microbiota symbiosis in disorders related to mitochondrial dysfunction.

Mitochondria are highly dynamic organelles that serve as bioenergetic and signaling hubs in eukaryotic cells, supporting relevant functions including calcium handling, regulation of apoptosis and oxidative stress, lipid biosynthesis, and, more importantly, production of the main cellular fuel − ATP. One of the peculiarities of mitochondria is that these organelles harbor their own genome, the mitochondrial DNA (mtDNA), which encodes 13 subunits of the electron transport chain that supports oxidative phosphorylation (OXPHOS) for ATP production[1]. Replication, transcription, and maintenance of mtDNA are controlled by the nuclear-encoded mitochondrial transcription factor A (TFAM)[2]. TFAM is essential for OXPHOS and mitochondrial biogenesis,

and germline TFAM deficiency is embryonically lethal. Therefore, generation of tissue-specific knockout mouse models has further illustrated the importance of mitochondria in different cell types[3–13]. Consistently, defects impinging on mitochondrial function precipitate the development of mitochondrial diseases, a group of inherited metabolic disorders showing severe muscular, cardiovascular, metabolic, and neurologic manifestations[14,15]. Although guidelines are available in the clinic to manage symptoms and complications of mitochondrial disease, limited therapeutic options are still accessible[16].

The intestine is a biological barrier regulated by an intimately attached epithelial cell layer that secretes mucus and antimicrobial

✉e-mail: mmittelbrunn@cbm.csic.es

peptides. Importantly, the collapse of this barrier is associated with increased morbidity and mortality in different species[6,17–19]. Beyond energy supply, mitochondrial metabolism plays a major role in gut homeostasis since it coordinates intestinal stem cell self-renewal and differentiation, cell death programs, smooth muscle cell contractility, and oxygen bioavailability in the lumen[20]. Accordingly, patients living with mitochondrial disorders usually suffer from chronic mucosal atrophy and gastrointestinal dysmotility with delayed emptying resulting in severe constipation[21]. In addition, the intestine harbors a highly diverse ecosystem of microorganisms, the gut microbiota, which lives in symbiosis with the host, providing pivotal metabolic pathways. Among others, short-chain fatty acids (SCFAs) synthesized by intestinal bacteria, such as acetate, propionate, and butyrate, strengthen the gut barrier and shape host physiology[22]. Mechanistically, SCFAs can trigger extracellular and intracellular signaling cascades as well as remodel the epigenetic landscape, therefore fine-tuning immunity, metabolism, and gene expression in the host[23–26]. Lately, research has shown that microbiota-derived butyrate regulates gene expression in the intestine of mice by controlling histone H3 butyrylation levels[27]. Given their pleiotropic effects, manipulation of SCFAs has been proposed as a therapeutic option for a myriad of inflammatory, cardiometabolic, and neurologic diseases[28]. While these results seem encouraging, the effectiveness of SCFAs in models of mitochondrial disease that display systemic multiorgan failure remains uncertain. In this work, we explore this line of intervention in a novel mouse model of mitochondrial dysfunction by inducible and ubiquitous deletion of TFAM, which mirrors the multimorbidity associated with mitochondrial diseases.

## Results

### Deletion of TFAM in adult mice induces systemic organ failure

To induce mitochondrial decline in adult mice, we crossed mice carrying loxP-flanked alleles of the gene encoding TFAM ($Tfam^{fl/fl}$) with mice expressing a tamoxifen-inducible Cre-ERT2 recombinase under the control of the ubiquitous $Ubc$ gene ($Ubc^{Cre-ERT2}$) to generate $Tfam^{fl/fl}Ubc^{Cre-ERT2}$ mice. This system enabled systemic and inducible knockdown of the $Tfam$ gene in adult mice (Fig. S1A), avoiding the reported embryonic lethality of germline TFAM deletion[2]. Consistently, we observed a remarkable drop in the levels of $Tfam$ transcripts in a myriad of tissues following tamoxifen administration (Fig. S1B). Over time, adult $Tfam^{fl/fl}Ubc^{Cre/ERT2+/-}$ mice (henceforth iTfamKO mice) manifested a progeroid phenotype starting 70 to 90 days after tamoxifen administration, with a shortened lifespan compared with age-matched control littermates (Fig. 1A, B).

In addition, 90 days after tamoxifen administration, these mice developed a multisystemic morbidity syndrome that encompasses a marked loss of body weight and lipodystrophy, showing an acute reduction of white and brown adipose tissues as well as smaller adipocytes (Fig. 1C, D and Fig. S1C, D). This was accompanied by diminished body temperature (Fig. S1E) and signs of glucose intolerance (Fig. 1E). iTfamKO mice also manifested hypogonadism, namely a reduced size of both testicles in males and ovaries in females (Fig. S1F). Regarding the cardiopulmonary system, iTfamKO mice displayed severe lung fibrosis (Fig. 1F) with increased lung weight, suggesting lung congestion or pulmonary edema (Fig. 1G). Histological evaluation of aortas revealed an increase in elastin dissections (Fig. 1H), in line with previous works of smooth muscle cell-specific deletion of $Tfam$[8]. Furthermore, analysis of skeletal muscle in iTfamKO mice uncovered a smaller diameter of muscle fibers and increased expression of growth differentiation factor 15 ($Gdf15$) and methylene tetrahydrofolate dehydrogenase 2 ($Mthfd2$), biomarkers of mitochondrial disease[29] which, together with a lower grip strength, were suggestive of severe sarcopenia (Fig. 1I-K). iTfamKO mice also exhibited signs of neurological disability, such as impaired locomotor coordination and activity in the rotarod and Open field tests (Fig. 1L, M and Fig. S1G), as well as reduced performance in the nest-building test (Fig. S1H). In addition,

the brain of iTfamKO mice was smaller (Fig. S1I) and harbored markers of neurodegeneration, such as increased neuronal lipofuscin aggregates (Fig. 1N), reduced levels of the postsynaptic marker PSD-95, and increased levels of phosphorylated Tau, as a consequence of successful depletion of $Tfam$ in the brain (Fig. S1J, K).

Importantly, iTfamKO mice displayed kidney failure with abnormally enlarged kidneys featuring signs of glomerulosclerosis and kidney tubule dilation (Fig. 1O), which was associated with polyuria and extremely clear urine, commonly found in diabetic conditions, as well as albuminuria (Fig. 1P and Fig. S1L). Furthermore, we observed that iTfamKO mice displayed heightened levels of interleukin (IL)−6 and tumor necrosis factor (TNF) in the serum (Fig. S1M). Similarly, the liver of iTfamKO mice showed enhanced expression of proinflammatory markers like IL-6 and TNF and downregulation of anti-inflammatory molecules like IL-10, together with strong expression of pro-fibrotic genes such as $Serpin-1$, $Ccn2$, and $Spp1$ (Fig. S1N). Of note, this was accompanied by increased expression of the senescence markers $Cdkn2a$ and $Cdkn1a$, respectively encoding the cyclin-dependent kinase inhibitors P16$^{INK4a}$ and P21$^{Waf/Cip1}$, in the liver of iTfamKO mice (Fig. S1N).

Lastly, the spleen and thymus of iTfamKO mice showed a substantial atrophy (Fig. 1Q, R), indicative of alterations in the hematopoietic system. Accordingly, hematological analysis of iTfamKO mice revealed decreased levels of parameters related to the number and function of erythrocytes (Fig. S2A), highlighting a profile of severe anemia (Fig. 1S). We also observed thrombocytosis and variations in leukocytes, including lymphopenia and neutrophilia, leading to a heightened neutrophil-to-lymphocyte ratio (NLR), a parameter predicting poor survival in multiple diseases[30], in iTfamKO mice compared with controls (Fig. 1T and Fig. S2B, C).

These data demonstrate that systemic ablation of $Tfam$ in adult mice drives systemic organ failure, including sarcopenia, cardiometabolic alterations, anemia, renal failure, and neurodegeneration, concomitant with the induction of a proinflammatory and senescent program in several tissues.

### Intestinal barrier integrity is disrupted in iTfamKO mice

Given the importance of mitochondrial fitness in maintaining intestinal barrier integrity[20], whose breakdown is associated with a plethora of pathological conditions[6,31], we sought to investigate the status of the small and large intestines in adult iTfamKO mice. We confirmed that the expression of $Tfam$ in both the ileum and the colon was notably diminished 90 days after the induction of the Cre recombinase expression (Fig. S3A). Accordingly, electron microscopy analysis uncovered that mitochondria in the ileum and the colon of iTfamKO mice were smaller and showed a remarkable increase in electrodense inclusion bodies (Fig. 2A, B), which were compatible with previously reported abnormal calcium phosphate aggregates due to a faulty handling of calcium[32]. Macroscopic examination of these organs revealed a significant increase in the length of the small intestine, but not in the colon, in iTfamKO mice compared with controls (Fig. S3B). Moreover, histological evaluation of the ileum and the colon showed an altered tissue architecture with notably smaller crypts (Fig. 2C). These alterations were concomitant with a decreased number of proliferative Ki67$^+$ cells in the crypts (Fig. S3C, D), and enhanced expression of the senescence marker P21$^{Waf/Cip1}$ in the ileum of iTfamKO mice compared with control littermates (Fig. S3E).

A crucial component of the intestinal barrier is the mucus layer provided by goblet cells, which prevents luminal bacteria from making direct contact with the epithelium, while serving as a habitat for commensal symbionts[33]. Although the number of goblet cells remains unaltered (Fig. S3F), the mucopolysaccharide content within these cells was remarkably increased in both the ileum and the colon of iTfamKO mice (Fig. 2D and Fig. S3G, H). This was accompanied by a decreased percentage of goblet cells secreting mucus into the

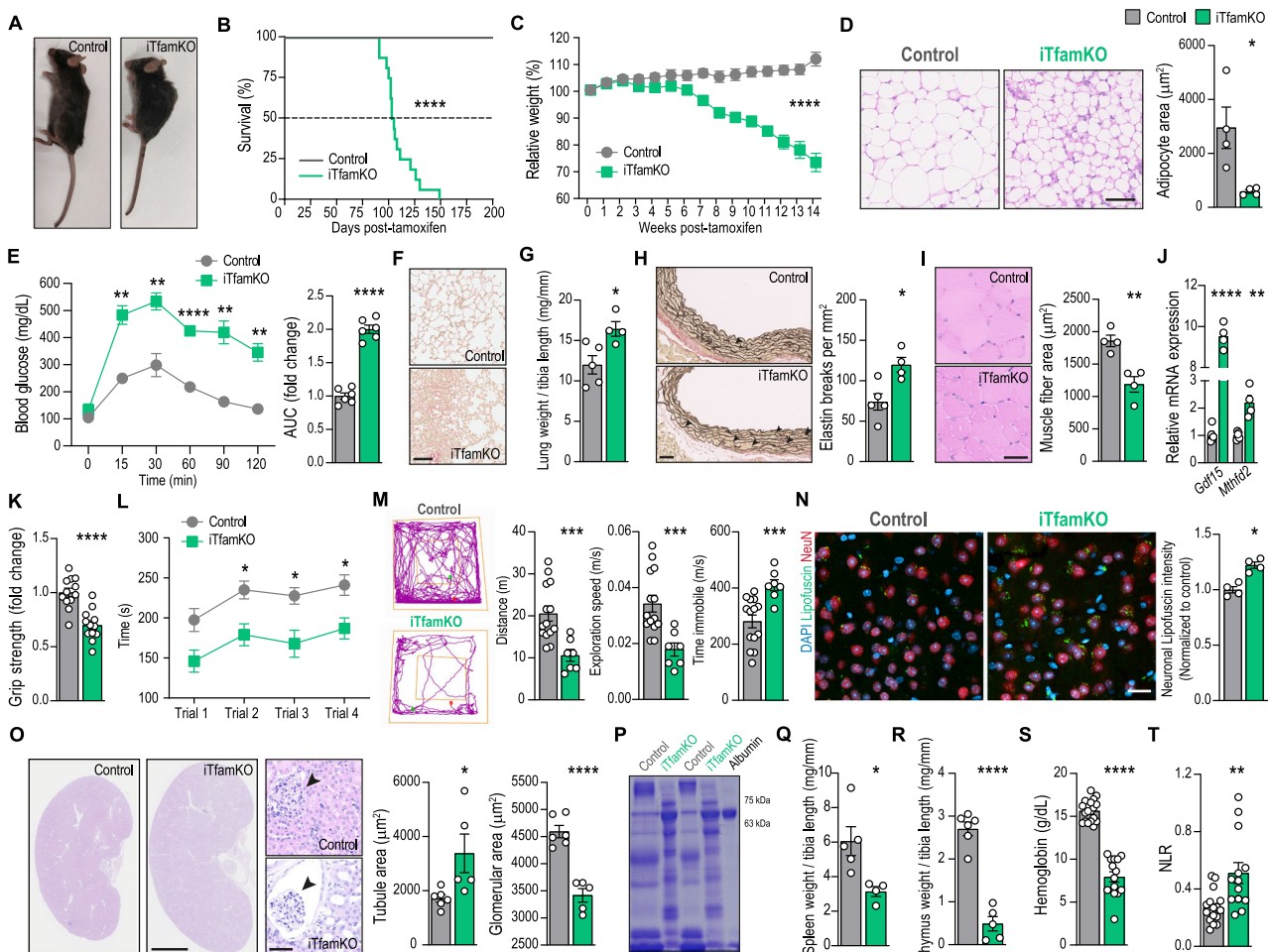

**Fig. 1 | Deletion of *Tfam* in adult mice induces systemic organ failure.**
**A** Representative picture of control and iTfamKO mice. **B** Kaplan–Meier survival curves (n = 7 for control, n = 16 for iTfamKO, ****P ≤ 0.0001). **C** Longitudinal assessment of relative body weight following tamoxifen administration (n = 17 for control, n = 15 for iTfamKO, ****P ≤ 0.0001). **D** Representative hematoxylin and eosin (H&E)–stained sections of gonadal white adipose tissue and quantification of adipocyte surface area (n = 4 for control, n = 4 for iTfamKO, *P = 0.0221). Scale bar: 50 µm. **E** Glucose tolerance tests and their respective area under the curve (AUC) (n = 6 for control, n = 6 for iTfamKO, **P = 0.0036 for t = 15 min, **P = 0.0094 for t = 30 min, ****P ≤ 0.0001 for t = 60 min, **P = 0.0081 for t = 90 min, **P = 0.0044 for t = 120 min, ****P ≤ 0.0001 for AUC). **F** Representative Sirius Red–stained sections of lung parenchyma. Scale bar: 100 µm. **G** Quantification of lung weight normalized to tibia length (n = 5 for control, n = 4 for iTfamKO, *P = 0.0217). **H** Representative elastic van Gieson (EVG)-stained sections of aorta and quantification of elastin breaks (n = 5 for control, n = 4 for iTfamKO, *P = 0.0132). Scale bar: 50 µm. **I** Representative H&E–stained sections of skeletal muscle and quantification of myofiber cross-sectional area (n = 4 for control, n = 4 for iTfamKO, **P = 0.0041). Scale bar: 50 µm. **J** Relative mRNA levels of the mitochondrial disease-associated genes *Gdf15* and *Mthfd2* in the skeletal muscle (n = 5 for control, n = 4 for iTfamKO, ****P ≤ 0.0001 for *Gdf15*, **P = 0.0018 for *Mthfd2*). **K** Forelimb grip strength test (n = 12 for control, n = 12 for iTfamKO, ****P ≤ 0.0001). **L** Rotarod test performance (n = 21 for control, n = 19 for iTfamKO, *P = 0.0121 for trial 2,

*P = 0.0191 for trial 3, *P = 0.021 for trial 4). **M** Open field test performance (n = 14 for control, n = 7 for iTfamKO, ***P = 0.0008 for distance, ***P = 0.0004 for exploration speed, ***P = 0.0005 for time immobile). **N** Representative image and quantification of lipofuscin particles in brain neurons (n = 4 for control, n = 4 for iTfamKO, *P = 0.0286). Scale bar: 20 µm. **O** Representative H&E–stained sections of kidney parenchyma and quantification of glomerular and tubule area (n = 6 for control, n = 5 for iTfamKO, *P = 0.0329 for tubule area, ****P ≤ 0.0001 for glomerular area). Scale bar: (whole kidney sections) 2.5 mm, (glomeruli) 50 µm. **P** Representative Coomassie-stained gel of urine samples. **Q** and **R** Quantification of (Q) spleen (n = 5 for control, n = 4 for iTfamKO, *P = 0.0211) and (R) thymus weight normalized to tibia length (n = 6 for control, n = 5 for iTfamKO, ****P ≤ 0.0001). **S** Concentration of hemoglobin (n = 18 for control, n = 14 for iTfamKO, ****P ≤ 0.0001). **T** Neutrophil-to-lymphocyte ratio (NLR) in the blood (n = 17 for control, n = 13 for iTfamKO, **P = 0.0015). All experiments were performed 90 days post-tamoxifen injection. Data are shown as means ± SEM, where each dot is a biological sample. P-values were determined by **B** log-rank (Mantel–Cox) test, **C** two-way analysis of variance (ANOVA), **D**, **G–K**, **M–O**, and **Q–S** unpaired two-tailed Student's t-test, **L** mixed-effects analysis with Šidák's multiple comparisons test, or **T** two-tailed Mann–Whitney U test. **E** P-values were determined by (curve) two-way ANOVA with Šidák's multiple comparisons test or (AUC) unpaired two-tailed Student's t-test. Source data and P-values for non-significant data are provided as a Source Data file.

intestinal lumen (Fig. 2E and Fig. S3I) and, accordingly, a notably thinner mucus layer (Fig. 2F and Fig. S3J), suggesting a defective goblet cell-mediated mucus secretion in this mouse model. To further assessed the physical integrity of the intestinal barrier, we evaluated the expression of genes encoding proteins involved in tight junctions (*Cldn1*, *Ocln*, and *Tjp1*), antimicrobial peptides (*Defa5*, *Defensin*, *Crypt1*, *Lyz1*, *Ang4*, and *Reg3g*) and participating in mucosa regeneration (*Tff3*), which showed an overall downregulation in the ileum, but not in the

colon, of iTfamKO mice compared with controls (Fig. 2G and Fig. S3K). This was translated into heightened intestinal permeability and translocation of bacteria to the periphery in iTfamKO mice, as confirmed by increased penetration of FITC-dextran after oral administration and elevated amounts of LPS-binding protein (LBP) in the serum, a surrogate marker of bacterial translocation (Fig. 2H, I).

We did not observe any traits of colitis, such as shortened colon or blood in the feces, at the steady state in iTfamKO mice. Instead, these

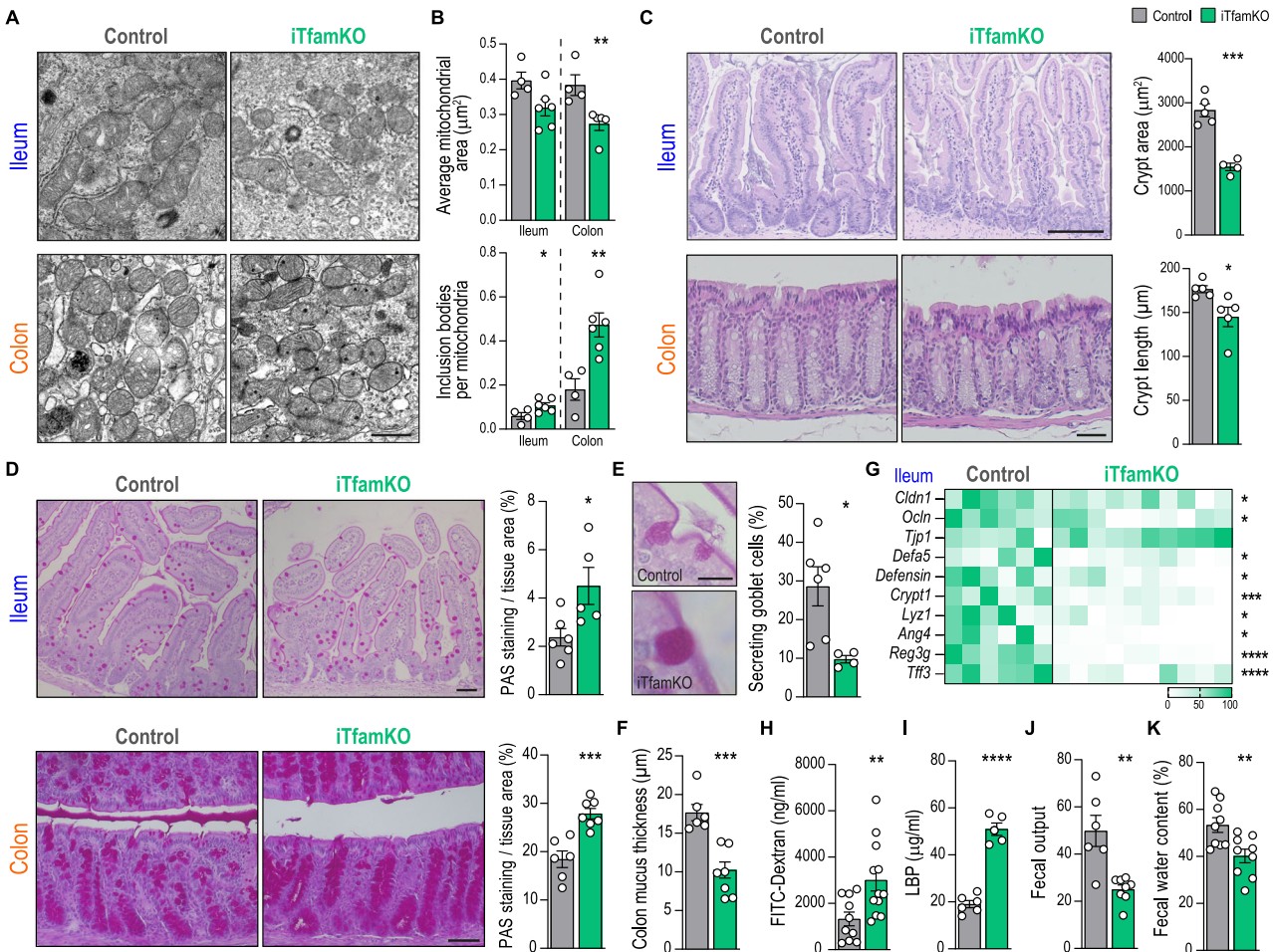

**Fig. 2 | iTfamKO mice display intestinal barrier disruption and signs of severe constipation.** **A** Representative electron microscopy images of the ileum and the colon epithelium in control and iTfamKO mice. Scale bar: 1 μm. **B** Quantification of mitochondrial area ($n = 4$ for control, $n = 6$ for iTfamKO, **$P = 0.0159$ for colon) and the number of electrodense inclusion bodies per mitochondria in the ileum and the colon epithelium ($n = 4$ for control, $n = 6$ for iTfamKO, *$P = 0.0367$ for ileum, **$P = 0.0056$ for colon). **C** Quantification of crypt dimension in the ileum ($n = 5$ for control, $n = 4$ for iTfamKO, ***$P = 0.0002$) and the colon ($n = 5$ for control, $n = 5$ for iTfamKO, *$P = 0.0281$). Scale bar: (ileum) 125 μm, (colon) 50 μm. **D** Representative Periodic Acid-Schiff (PAS)−stained sections and quantification of stained area in the ileum ($n = 6$ for control, $n = 5$ for iTfamKO, *$P = 0.026$) and the colon ($n = 6$ for control, $n = 7$ for iTfamKO, ***$P = 0.0006$). Scale bar: 50 μm. **E** Representative image and quantification of the percentage of secreting goblet cells in the ileum ($n = 6$ for control, $n = 4$ for iTfamKO, *$P = 0.0182$). Scale bar: 10 μm. **F** Quantification of mucus layer thickness in the colon ($n = 6$ for control, $n = 7$ for iTfamKO, ***$P = 0.0004$).

**G** Heatmap depicting normalized values of expression levels of genes associated with tight junctions (*Cldn1*, *Ocln*, *Tjp1*), antimicrobial peptides (*Defa5*, *Defensin*, *Crypt1*, *Lyz1*, *Ang4*, *Reg3g*), and barrier function (*Tff3*) in the ileum assessed by qPCR ($n = 6$ for control, $n = 10$ for iTfamKO, *$P = 0.0424$ for *Cldn1*, *$P = 0.0262$ for *Ocln*, *$P = 0.0283$ for *Defa5*, *$P = 0.0378$ for *Defensin*, ***$P = 0.0008$ for *Crypt1*, *$P = 0.0181$ for *Lyz1*, *$P = 0.0141$ for *Ang4*, ****$P \leq 0.0001$ for *Reg3g* and *Tff3*). **H** Concentration of FITC-dextran in the serum ($n = 10$ for control, $n = 12$ for iTfamKO, **$P = 0.0093$). **I** Levels of LPS-binding protein (LBP) in the serum ($n = 6$ for control, $n = 5$ for iTfamKO, ****$P \leq 0.0001$). **J** Quantification of the number of feces defecated overnight ($n = 6$ for control, $n = 8$ for iTfamKO, **$P = 0.0015$). **K** Percentage of water in the feces ($n = 9$ for control, $n = 9$ for iTfamKO, **$P = 0.0071$). All experiments were performed 90 days post-tamoxifen injection. Data are shown as means ± SEM, where each dot is a biological sample. *P*-values were determined by **B**−**K** unpaired two-tailed Student's *t*-test. Source data and *P*-values for non-significant data are provided as a Source Data file.

mice exhibited marked signs of constipation, as demonstrated by a decrease in the frequency of defecated stool samples (Fig. 2J), together with a substantially reduced weight and percentage of water content in feces (Fig. 2K and Fig. S3L), suggesting defects in gut peristalsis and water absorption.

## Systemic deficiency of *Tfam* leads to bacterial dysbiosis and reduced SCFA levels

Metabolic perturbations in the intestine are associated with pathogenic shifts in the gut microbiota[34,35]. To characterize the composition of the bacterial communities in the intestine of iTfamKO mice, we performed *16S* rRNA gene sequencing in samples of the terminal ileum and the colon in these mice 90 post-tamoxifen administration. Analyses of α-diversity indicated a reduction in parameters such as

Shannon and observed species (Sobs) indexes in the ileal but not in the colonic microbiota (Fig. 3A, B). Moreover, β-diversity analyses by nonmetric multidimensional scaling (NMDS) denoted a remarkably distinct configuration of the gut microbiota in the colon of iTfamKO mice compared with control littermates (Fig. 3C, D). Compositional analysis in both intestinal compartments suggested a striking reduction in bacteria belonging to the Clostridiales order, such as *Lachnospiraceae* and *Ruminococcaceae* families in the ileal and colonic microbiome of iTfamKO mice when compared with controls (Fig. 3E). These changes were accompanied by an expansion of bacteria from the Bacillales order, including members of the *Staphylococcaceae* family in both the ileum and the colon of this mouse model (Fig. 3E).

To delve into the *16S* rRNA metagenomic data, we performed a predictive functional analysis of the gut microbiome. Notably,

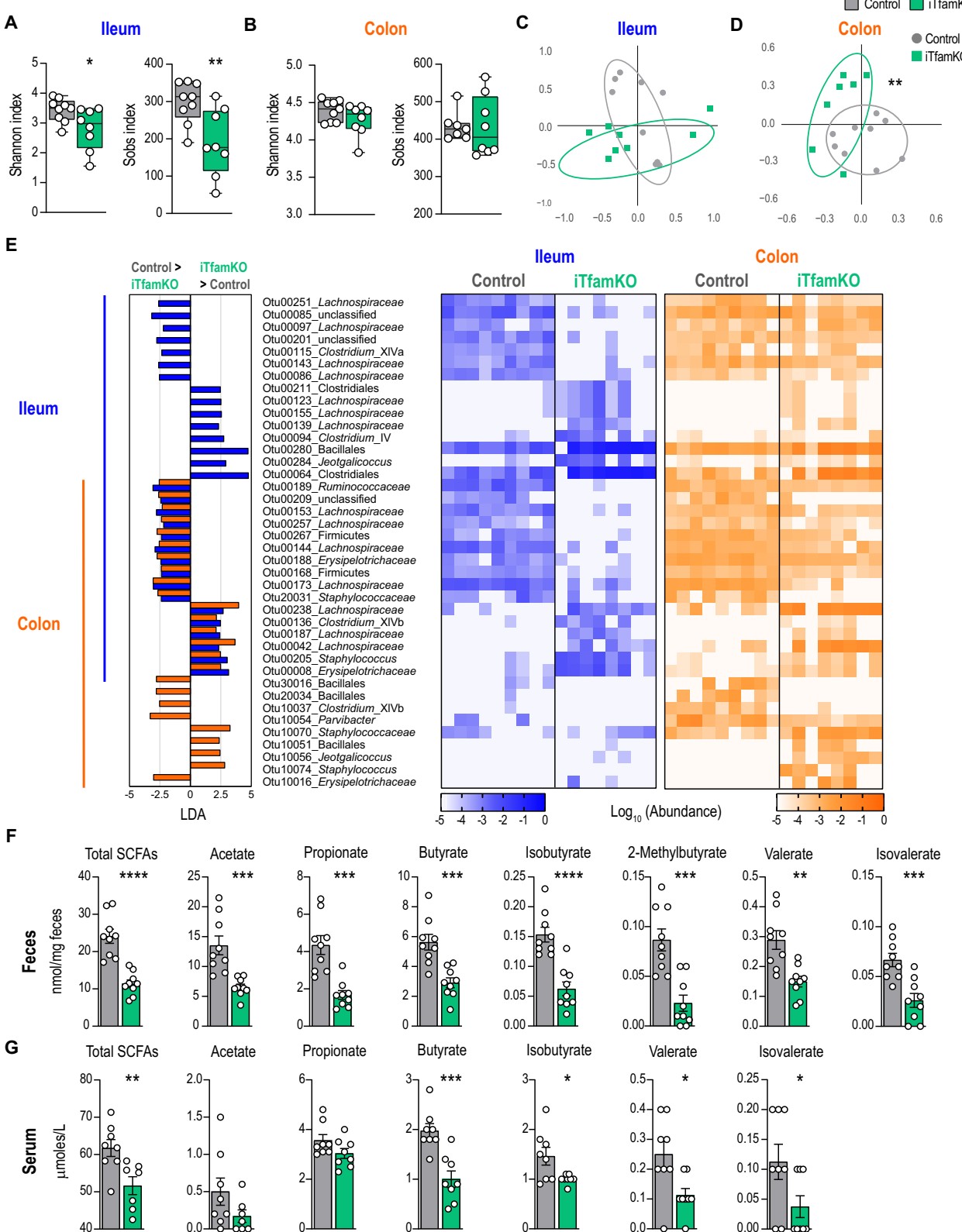

sequencing profiles of the ileum-dwelling microbiota unveiled a reduction in biochemical pathways related to carbohydrate degradation, for instance, fucose, D-glucarate, and rhamnose in iTfamKO mice (Fig. S4). Polysaccharide utilization by the gut microbiota leads to the balanced production of health-promoting metabolites such as SCFAs[36]. To examine whether the observed compositional and metabolic changes of the gut microbiota impinged on the production of

SCFAs, we quantified these molecular species in fecal and serum samples of mice using liquid chromatography-mass spectrometry (LC-MS). Remarkably, we observed a significant reduction in the concentration of total SCFAs, as well as in each one of them (i.e., acetate, propionate, butyrate, isobutyrate, 2-methylbutyrate, valerate, and isovalerate) in the feces of these mice (Fig. 3F). This correlated with downregulation of most SCFAs in cecal and colonic, but not in ileal,

**Fig. 3 | Systemic deficiency of *Tfam* leads to bacterial dysbiosis and reduced SCFA production.** Shannon index and observed species index (Sobs) parameters of α-diversity in (**A**) the ileum ($n = 9$ for control, $n = 8$ for iTfamKO, *$P = 0.0365$ for Shannon index, **$P = 0.0089$ for Sobs) and (**B**) colon-resident microbiota ($n = 9$ for control, $n = 8$ for iTfamKO) of control and iTfamKO mice. Non-metric multi-dimensional scaling (NMDS) plots of β-diversity values (θYC indexes) in (**C**) the ileum ($n = 9$ for control, $n = 8$ for iTfamKO) and (**D**) colon-resident microbiota ($n = 9$ for control, $n = 7$ for iTfamKO, **$P = 0.0074$). **E** Left: differentially abundant operational taxonomic units (OTUs) depicted with linear discriminant analysis (LDA) values of linear discriminant effect size (LEfSe, $P < 0.05$; FDR, $Q < 0.05$; | SNR| > 0.5; |LDA| > 2; |fold change| > 10; maximal abundance > 0.001) comparing ileal and colonic microbiota in control versus iTfamKO mice. Right: heatmap depicting abundance values. Quantification of short-chain fatty acids (SCFAs) in (**F**)

the feces ($n = 9$ for control, $n = 9$ for iTfamKO, ****$P \le 0.0001$ for total SCFA, ***$P = 0.0006$ for acetate, ***$P = 0.0002$ for propionate, ***$P = 0.0004$ for butyrate, ****$P \le 0.0001$ for isobutyrate, ***$P = 0.0003$ for 2-methylbutyrate, **$P = 0.0013$ for valerate, ***$P = 0.0007$ for isovalerate) and **G** the serum ($n = 8$ for control, $n = 8$ for iTfamKO, **$P = 0.0092$ for total SCFA, ***$P = 0.0006$ for butyrate, *$P = 0.0236$ for isobutyrate, *$P = 0.0184$ for valerate, *$P = 0.0486$ for isovalerate). All experiments were performed 90 days post-tamoxifen injection. Data are shown as means ± SEM, where each dot is a biological sample. Box-and-whisker plots represent the inter-quartile range between the first and third quartiles (25th and 75th percentiles, respectively), the median, and the maximal and minimal values. $P$-values were determined by **A**, **B**, **F**, and **G** unpaired two-tailed Student's $t$-test, **C** and **D** permutational multivariate analysis of variance (PERMANOVA), or **E** LEfSe. Source data and $P$-values for non-significant data are provided as a Source Data file.

contents (Fig. S5A). Supporting that these alterations also occur systemically, we observed a significant reduction of total and most individual SCFA species, with the exception of acetate and propionate, in the serum of iTfamKO mice when compared with control littermates (Fig. 3G).

In an effort to shed light on which bacteria could be involved in this drop of SCFAs, we performed Spearman correlation analyses between the ileum and colon-resident microbiota and the concentration of SCFA species in the same samples. Focusing on the operational taxonomic units (OTUs) that decreased in the gut microbiota of iTfamKO mice, the results unveiled ten OTUs that positively correlated with the reduced levels of these molecules (Fig. S5B). Among them, we found members belonging to the *Ruminococcaceae* and *Lachnospiraceae* families, which represent crucial SCFA-producing bacteria in the gut microbiota.

Altogether, our findings suggest that deletion of *Tfam* leads to mitochondrial alterations in the intestine, resulting in the rupture of the intestinal barrier, gut dysmotility, and bacterial dysbiosis that is associated with a marked reduction in SCFA production.

## mtDNA-mutator mice display gut dysbiosis and reduced levels of butyrate

We wondered whether another mouse model of mitochondrial deficiency showing premature multimorbidity also exhibited alterations in intestinal barrier integrity and gut microbiota composition. For that, we examined *PolgA*$^{D257A/D257A}$ (*Polg*$^{mut}$) or mtDNA-mutator mice that carry a deficient proof-reading version of the gene encoding the mtDNA polymerase gamma, therefore leading to a progressive accumulation of somatic mutations in the mtDNA that trigger premature multimorbidity[37,38]. We analyzed *Polg*$^{mut}$ mice at 50 weeks of age, once they show signs of multimorbidity, compared with age-matched control mice (Fig. 4A).

Notably, *Polg*$^{mut}$ mice displayed intestinal barrier dysfunction with 2-fold heightened levels of LBP in the serum compared with control littermates (Fig. 4B). This was accompanied by downregulated expression of genes encoding antimicrobial peptides, such as *Defa5*, *Defensin*, *Crypt*, and *Ang4* in the ileum, and reduced expression of genes participating in tight junction architecture like *Ocln* and *Tjp1* in the colon of *Polg*$^{mut}$ mice compared with controls (Fig. 4C). Analyses of the fecal microbiota in these mice unveiled a marked reduction in α-diversity parameters such as Shannon index and Shannon evenness in *Polg*$^{mut}$ mice (Fig. 4D), suggesting loss of bacterial species in this mouse model. In addition, analyses of β-diversity by NMDS demonstrated a substantially different configuration of the fecal microbiota (Fig. 4E). Similar to that observed in iTfamKO mice, LC-MS quantification of SCFA species in the feces of *Polg*$^{mut}$ mice uncovered decreased levels of butyrate in these mice when compared to control mice (Fig. 4F). Remarkably, predictive functional analysis of the *16S* rRNA metagenomic data showed downregulation of biochemical pathways related to carbohydrate utilization (Fig. S6). In addition, evaluation of predictive metabolic pathways shared by iTfamKO and *Polg*$^{mut}$ mice showed downregulation in

pathways linked to the degradation of aromatic compounds (Fig. S7), which are usually metabolized into acetyl-Coenzyme A (acetyl-CoA) − a substrate for butyrate biosynthesis in intestinal bacteria.

Evaluation of the relative abundance of bacteria in the feces of these mice showed an overall reduction in bacteria belonging to the Clostridiales order, such as the *Lachnospiraceae* and *Ruminococcaceae* families, and belonging to Bacteroidales, including the *Porphyromonadaceae* family. We found an abnormal expansion of *Bacteroidaceae* and some members of *Enterobacteriaceae* in *Polg*$^{mut}$ mice compared with control littermates (Fig. 4G). Spearman correlation analysis between differentially abundant OTUs and butyrate levels revealed seven OTUs that were diminished in these mutant mice and positively correlated with the decreased levels of butyrate. These OTUs corresponded to well-known butyrate-producing members of the *Lachnospiraceae*, *Ruminococcaceae*, and *Porphyromonadaceae* families (Fig. 4H).

Thereby, our findings reveal that an alternative mouse model of multimorbidity due to mitochondrial dysfunction manifests disruption of the intestinal barrier integrity as well as gut dysbiosis, featured by the loss of butyrate-producing taxa.

## Butyrate supplementation improves systemic organ failure, extending health and lifespan in iTfamKO mice

Given the intestinal dysbiosis found in iTfamKO mice, we investigated the contribution of the gut microbiota to the multimorbidity phenotype of these mice. Thus, we designed a strategy to replace the gut microbiota with fecal microbiota transplantation (FMT) assays. In brief, we transferred fecal microbiota from either control or iTfamKO donor mice to iTfamKO recipient mice (denoted as iTfamKO FMT$_{Control}$ and iTfamKO FMT$_{KO}$, respectively) starting 30 days after tamoxifen administration (Fig. 5A). iTfamKO FMT$_{Control}$ mice showed slightly delayed loss of body weight and increased muscle strength when compared with iTfamKO FMT$_{KO}$ mice (Fig. 5B, C), with no changes in the levels of LBP (Fig. 5D). Moreover, water content in the feces was recovered to the levels of control mice in iTfamKO FMT$_{Control}$ mice (Fig. 5E), suggesting improvement in signs of constipation. On top of this, we found that the levels of some SCFAs, such as butyrate and, to a lesser extent, isobutyrate, 2-methylbutyrate, and isovalerate, were restored in iTfamKO mice receiving microbiota from healthy control mice (Fig. 5F). Notably, this was associated with an extension in the maximum lifespan by 70% (from 115 to 196 days) in iTfamKO FMT$_{Control}$ mice compared with iTfamKO FMT$_{KO}$ mice (Fig. 5G).

Since FMT from healthy control mice rescued the levels of butyrate in iTfamKO recipient mice, which is associated with improvement in healthspan and mortality, we further explored the therapeutic potential of restoring butyrate levels in this mouse model. For that purpose, iTfamKO mice were fed a diet supplemented with 10% tributyrin (TB), a butyrate precursor in the form of a triacyl-glycerol ester of butyric acids, starting 30 days after tamoxifen administration (Fig. 6A). iTfamKO mice fed a TB-supplemented diet showed increased

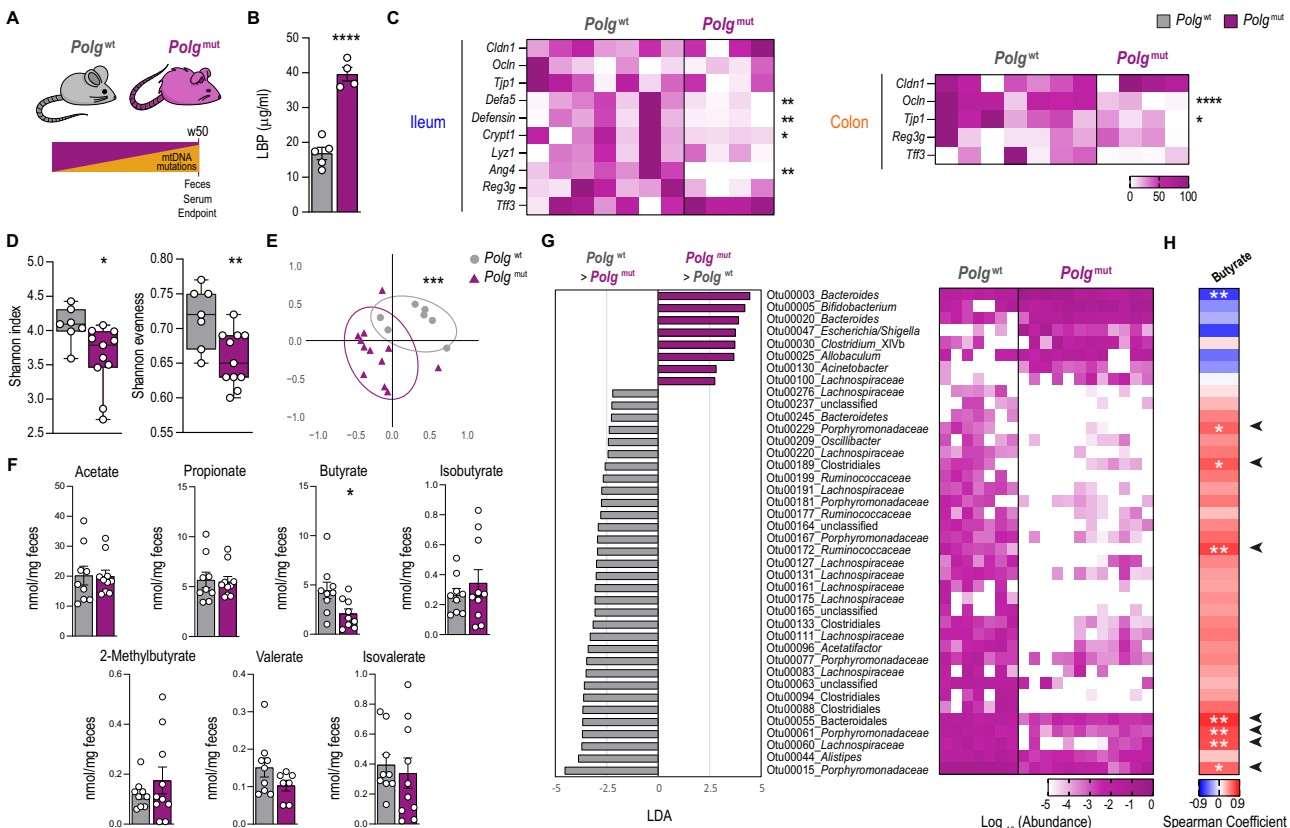

**Fig. 4 | mtDNA-mutator mice show bacterial dysbiosis and reduced levels of butyrate. A** Scheme of 50-week-old *Polg*wt and *Polg*mut mice. **B** Levels of LPS-binding protein (LBP) in the serum (*n* = 5 for *Polg*wt, *n* = 4 for *Polg*mut, ****\**P* ≤ 0.0001). **C** Heatmap depicting normalized values of expression levels of genes associated with tight junctions (*Cldn1, Ocln, Tjp1*), antimicrobial peptides (*Defa5, Defensin, Crypt1, Lyz1, Ang4, Reg3g*), and barrier function (*Tff3*) in the ileum (*n* = 7 for *Polg*wt, *n* = 4 for *Polg*mut, **\**P* = 0.0095 for *Defa5*, **\**P* = 0.0095 for *Defensin*, *\*P* = 0.05 for *Crypt1*, **\**P* = 0.0095 for *Ang4*) and the colon (*n* = 7 for *Polg*wt, *n* = 4 for *Polg*mut, ****\**P* ≤ 0.0001 for *Ocln*, *\*P* = 0.0324 for *Tjp1*) assessed by qPCR. **D** Shannon index and Shannon evenness parameters of α-diversity in fecal microbiota (*n* = 7 for *Polg*wt, *n* = 11 for *Polg*mut, *\*P* = 0.0262 for Shannon index, **\**P* = 0.0077 for Shannon evenness). **E** Non-metric multidimensional scaling (NMDS) plots of β-diversity values (θYC indexes) in fecal microbiota (*n* = 7 for *Polg*wt, *n* = 12 for *Polg*mut, ***\**P* = 0.0001). **F** Quantification of short-chain fatty acid species in the feces (*n* = 9

for *Polg*wt, *n* = 10 for *Polg*mut, *\*P* = 0.0266 for butyrate). **G** Left: differentially abundant operational taxonomic units (OTUs) depicted with linear discriminant analysis (LDA) values of linear discriminant effect size (LEfSe, *P* < 0.05; FDR, *Q* < 0.05; |SNR| > 0.5; |LDA| > 1; |fold change| > 5; maximal abundance > 0.001) comparing fecal microbiota in *Polg*wt versus *Polg*mut mice. Right: heatmap depicting abundance values. **H** Heatmap depicting Spearman's rank correlation coefficients between differentially abundant OTUs and the concentration of butyrate in the feces. Data are shown as means ± SEM, where each dot is a biological sample. Box-and-whisker plots represent the interquartile range between the first and third quartiles (25th and 75th percentiles, respectively), the median, and the maximal and minimal values. *P*-values were determined by **B**–**D**, and **F** unpaired two-tailed Student's *t*-test, **E** permutational multivariate analysis of variance (PERMANOVA), **G** LEfSe, or **H** using the OTU association command of mothur (v.1.40.5) software package. Source data and *P*-values for non-significant data are provided as a Source Data file.

levels of butyrate in the feces compared with iTfamKO mice fed a standard diet (Fig. 6B and Fig. S8A). Butyrate retrieval was associated with a delayed loss of body weight (Fig. 6C), and notably enhanced muscle strength assessed by grip test, which was accompanied by diminished expression of the mitochondrial disease-associated genes *Gdf15* and *Mthfd2* in the skeletal muscle (Fig. 6D, E). Furthermore, iTfamKO mice fed a TB-supplemented diet showed improved glucose handling by glucose tolerance tests, and restored levels of fasting glucose almost to the levels of control mice (Fig. 6F, G). Furthermore, quantification of albumin levels in the urine indicated improvement in the function of the kidney in iTfamKO mice fed a TB-supplemented diet (Fig. 6H). TB supplementation affected neither spleen size nor hematological parameters in treated iTfamKO mice (Fig. S8B, C). Importantly, TB administration remarkably extended median lifespan by ~ 25% (from 98 to 123 days) and maximum lifespan by more than 75% (from 106 to 186 days) in iTfamKO mice fed a TB-supplemented diet when compared with those receiving a standard diet (Fig. 6I).

Taken together, our findings support the notion that interventions to restore butyrate levels delay signs of multimorbidity and extend the lifespan in iTfamKO mice.

## Butyrate retrieval restores epigenetic histone acylation marks in the intestine of iTfamKO mice

Gut microbiota-derived butyrate has been shown to remodel the epigenetic landscape of intestinal cells through histone H3 acylation[24,27]. To elucidate the molecular mechanisms underlying the beneficial effects of TB supplementation, we analyzed the levels of acetylation and butyrylation on lysines 9 and 27 of histone H3 in the ileum by Western Blotting. Compared with control mice, iTfamKO mice exhibited a notable reduction in the levels of H3K9ac, H3K9bu, and H3K27bu in the ileum (Fig. 7A). Confirming the crucial role of microbiota-derived SCFAs in the acquisition of these histone modifications, we observed similar results in microbiota-depleted control mice after treatment with a cocktail of broad-spectrum antibiotics (Abx) (Fig. 7A), which showed a nearly complete loss of SCFAs compared with controls receiving vehicle (Fig. 7B).

We explored the potential of TB in restoring these histone modifications in iTfamKO mice. Remarkably, TB supplementation restored the levels of H3K9ac, H3K9bu, and H3K27bu in the ileum of iTfamKO mice when compared with those receiving a standard diet (Fig. 7C). To investigate how TB affected the transcriptomic landscape in the ileum,

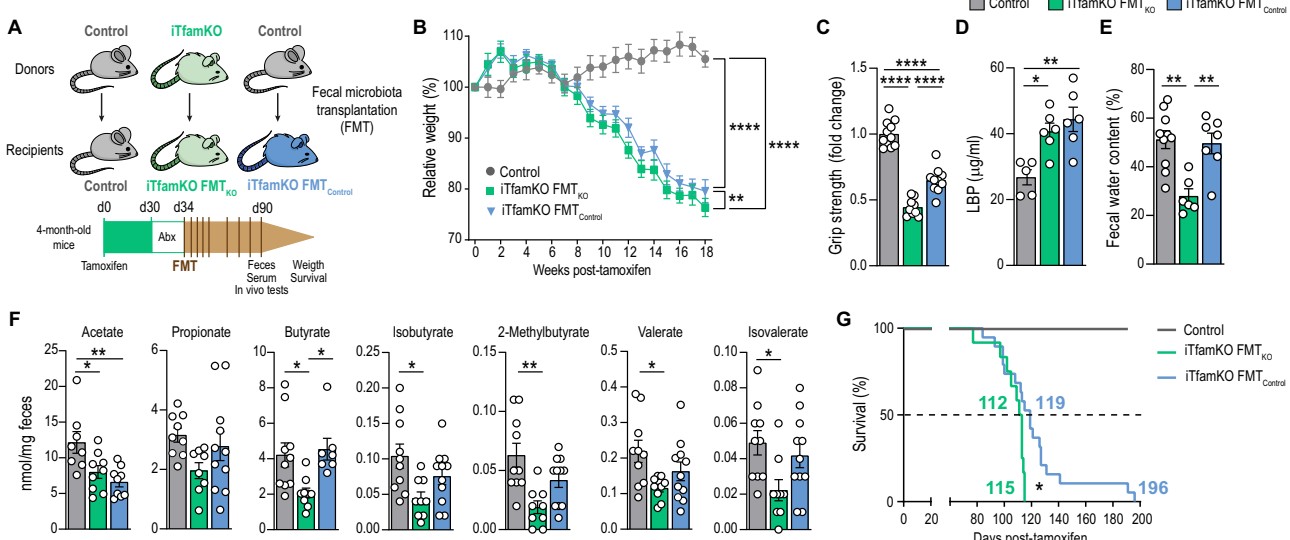

**Fig. 5 | Transplantation of control microbiota ameliorates signs of multi-morbidity in iTfamKO mice. A** Experimental design of fecal microbiota transplantation (FMT) assays from donor control or iTfamKO mice into recipient iTfamKO mice (iTfamKO FMT$_{Control}$ or iTfamKO FMT$_{KO}$ mice, respectively). **B** Longitudinal assessment of body weight relative to tamoxifen administration ($n = 19$ for control, $n = 21$ for FMT$_{KO}$, $n = 23$ for FMT$_{Control}$, ****$P \leq 0.0001$ for control versus FMT$_{KO}$ and control versus FMT$_{Control}$, **$P = 0.0051$ for FMT$_{KO}$ versus FMT$_{Control}$). **C** Forelimb grip strength analysis ($n = 11$ for control, $n = 9$ for FMT$_{KO}$, $n = 10$ for FMT$_{Control}$, ****$P \leq 0.0001$). **D** Levels of LPS-binding protein (LBP) in the serum ($n = 5$ for control, $n = 6$ for FMT$_{KO}$, $n = 6$ for FMT$_{Control}$, *$P = 0.0156$ for control versus FMT$_{KO}$, **$P = 0.0029$ for control versus FMT$_{Control}$). **E** Percentage of water in the feces ($n = 10$ for control, $n = 6$ for FMT$_{KO}$, $n = 7$ for FMT$_{Control}$, **$P = 0.0011$ for

control versus FMT$_{KO}$, **$P = 0.0041$ for FMT$_{KO}$ versus FMT$_{Control}$). **F** Quantification of short-chain fatty acid species in the feces ($n = 8$ for control, $n = 9$ for FMT$_{KO}$, $n = 9$ for FMT$_{Control}$; acetate: *$P = 0.0296$ for control versus FMT$_{KO}$, **$P = 0.0035$ for control versus FMT$_{Control}$; butyrate: *$P = 0.0238$ for control versus FMT$_{KO}$, *$P = 0.0176$ for FMT$_{KO}$ versus FMT$_{Control}$; *$P = 0.0118$ for isobutyrate; **$P = 0.0012$ for 2-methylbutyrate; *$P = 0.0229$ for valerate; *$P = 0.0266$ for isovalerate). **G** Kaplan–Meier survival curves ($n = 9$ for control, $n = 12$ for FMT$_{KO}$, $n = 19$ for FMT$_{Control}$, *$P = 0.0133$ for FMT$_{KO}$ versus FMT$_{Control}$). Data are shown as means ± SEM, where each dot is a biological sample. $P$-values were determined by **B** two-way analysis of variance (ANOVA) with Tukey's multiple comparisons test, **C**–**F** one-way ANOVA with Tukey's multiple comparisons test, or **G** log-rank (Mantel–Cox) test. Source data and $P$-values for non-significant data are provided as a Source Data file.

we performed bulk RNA-sequencing analysis in samples from control, iTfamKO, and TB-supplemented iTfamKO mice. Hierarchical clustering of differentially expressed genes showed that control and TB-supplemented iTfamKO mice grouped together (Fig. 7D). This suggested that TB partially restored gene expression alterations in the ileum of these mice, with upregulation of genes involved in the structural integrity of the mucosa (*Vill, Krt80, Sh3pxd2b, Loxl1, Matn2*) and regulation of the immune response (*FoxP3, Ctla4, Ccr3, Ccl22*) (Fig. S9), which are crucial for the correct performance of the intestinal barrier. Butyrylation of histone H3 has been recently linked to the transcriptional regulation of genes involved in relevant pathways for intestinal homeostasis, such as intercellular junctional complexes and oxidative stress[27]. Therefore, we examined the expression of genes previously published to be dependent on the H3K27bu mark[27] (Fig. 7E). Focusing on the genes whose expression was repressed in the ileum of iTfamKO mice compared with controls, we found that TB upregulated genes involved in cell-to-cell junction architecture (*Tjp2, Cdh2, Dsp*) and cytoskeleton organization (*Nck2, Rhpn2, Dyrk1a, Krt78*), which are relevant for the regulation of the physical barrier of the intestine (Fig. 7F and Fig. S10). Furthermore, we observed increased expression of genes participating in the oxidative stress response (*Jun, Src*) as well as in cellular proliferation and tissue morphogenesis, with special emphasis on muscle cell functionality (*Tgfbr2, Zfand5*) that may play a role in gut peristalsis (Fig. 7F). Thus, butyrate supplementation restores epigenetic histone modifications in the intestine of iTfamKO mice correcting the expression of some genes involved in intestinal homeostasis.

Overall, our results indicate that intestinal barrier disruption and gut dysbiosis are hallmarks of mitochondrial dysfunction, leading to reduced SCFA production. Therefore, restoring butyrate levels alleviates systemic organ failure and remarkably extends lifespan in iTfamKO mice.

## Discussion

Disruption of intestinal homeostasis, comprising increased intestinal permeability and microbial dysbiosis, has been increasingly associated with a wide spectrum of non-communicable pathologies, including inflammatory, metabolic, cardiovascular, and neuropsychiatric disorders[6,31]. Although the precise role of gut dysbiosis in pathology remains scarcely understood, its negative impact on the host has been attributed to its dysregulated metabolic byproducts. SCFAs, key metabolites produced by the gut microbiota through fermentation of dietary and host carbohydrates and proteins, play a central role in safeguarding intestinal barrier integrity and host physiology. Given its local and systemic beneficial effects, SCFAs have emerged as promising therapeutic agents for a plethora of inflammatory, oncologic, cardiometabolic, and neurological conditions[28]. Thereby, the interplay between the host and its commensal microbiota is instrumental to preserving mutualism and ensuring health.

In this study, we demonstrate that two distinct mouse models of systemic mitochondrial dysfunction − iTfamKO and mtDNA-mutator mice − exhibit intestinal barrier disruption accompanied by profound bacterial dysbiosis, which is featured by a marked reduction in microbial diversity and loss of commensal symbionts. We observed that mitochondrial failure is associated with gut hyperpermeability in both iTfamKO and mtDNA-mutator mice. Previous reports showed that adult-onset ablation of *Tfam* specifically in the intestinal epithelium impinges on intestinal stem cell (ISC) dynamics, blunting enterocyte maturation[39]. mtDNA-mutator mice also show intestinal crypt atrophy and decreased number and activity of proliferative ISCs due to NAD$^+$ depletion and subsequent integrated stress response activation[40,41], which might participate in the breakdown of the intestinal physical barrier in our mouse models. In addition to the epithelium, *Tfam* depletion also affects smooth muscle cells in the intestinal mucosa of iTfamKO mice, which critically disturbs gut

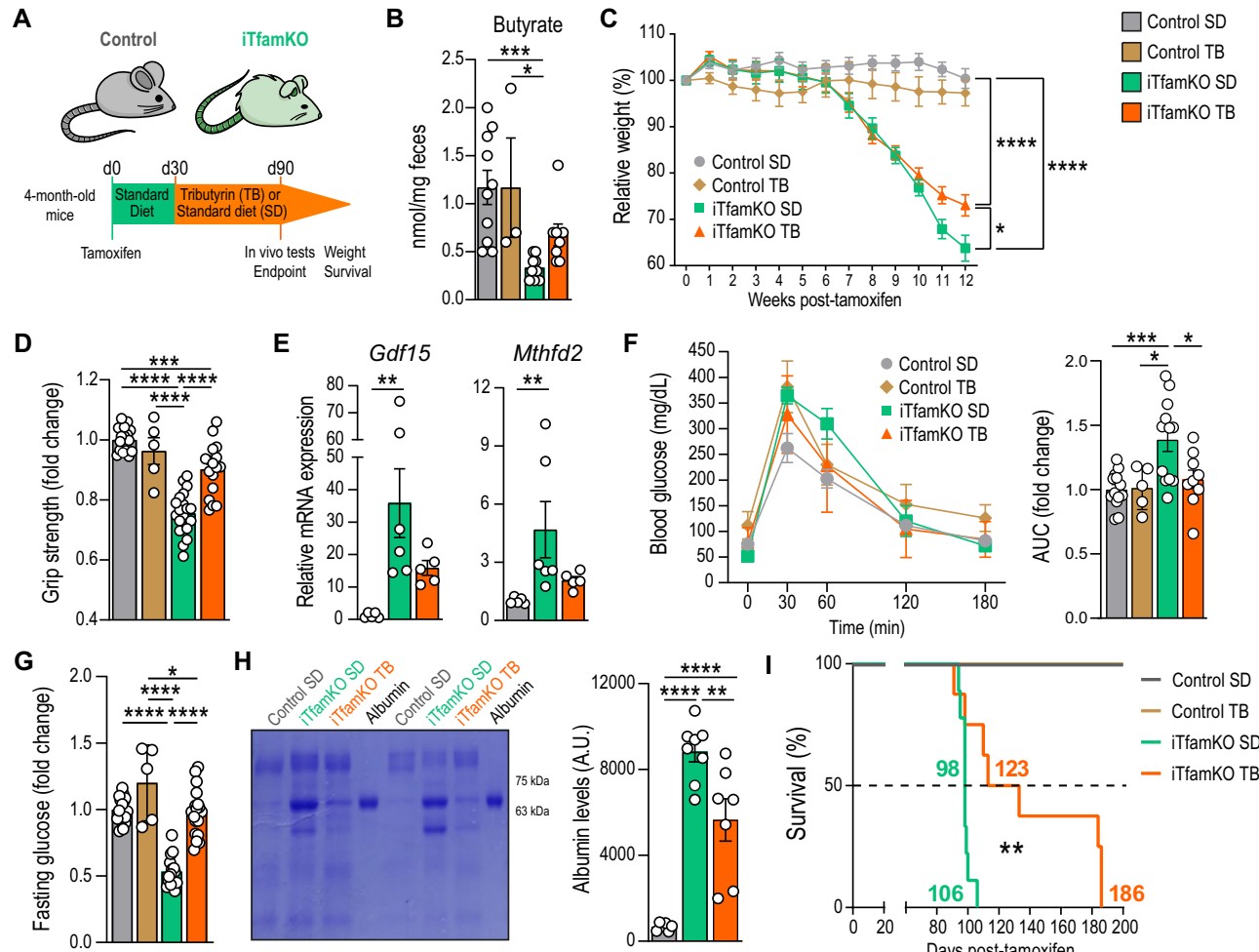

Fig. 6 | **Tributyrin supplementation extends health and lifespan in iTfamKO mice. A** Experimental design of control and iTfamKO mice fed either a standard diet (SD) or a 10% tributyrin-supplemented diet (TB). **B** Quantification of butyrate in the feces ($n = 10$ for control SD, $n = 3$ for control TB, $n = 9$ for iTfamKO SD, $n = 8$ for iTfamKO TB, ***$P = 0.0005$ for control SD versus iTfamKO SD, *$P = 0.04$ for iTfamKO SD versus Control TB). **C** Longitudinal assessment of body weight relative to tamoxifen administration ($n = 19$ for control SD, $n = 5$ for control TB, $n = 21$ for iTfamKO SD, $n = 19$ for iTfamKO TB, ****$P \leq 0.0001$ for control SD versus iTfamKO SD and control SD versus iTfamKO TB, *$P = 0.03$ for iTfamKO SD versus iTfamKO TB). **D** Forelimb grip strength analysis ($n = 18$ for control SD, $n = 5$ for control TB, $n = 20$ for iTfamKO SD, $n = 17$ for iTfamKO TB, ****$P \leq 0.0001$ for control SD versus iTfamKO SD, iTfamKO SD versus iTfamKO TB, and iTfamKO SD versus control TB, ***$P = 0.0007$ for control SD versus iTfamKO TB). **E** Relative mRNA levels of the mitochondrial disease-associated genes *Gdf15* and *Mthfd2* in the skeletal muscle ($n = 5$ for control SD, $n = 6$ for iTfamKO SD, $n = 5$ for iTfamKO TB; *Gdf15*: **$P = 0.0036$ for control SD versus iTfamKO SD; *Mthfd2*: **$P = 0.0029$ for control SD versus iTfamKO SD). **F** Glucose tolerance test and its respective area under the curve (AUC) quantification ($n = 13$ for control SD, $n = 5$ for control TB, $n = 12$ for iTfamKO SD, $n = 9$ for iTfamKO TB; AUC: ***$P = 0.0009$ for control SD versus

iTfamKO SD, *$P = 0.0199$ for control TB versus iTfamKO SD, *$P = 0.0205$ for iTfamKO SD versus iTfamKO TB). **G** Quantification of fasting glucose levels ($n = 13$ for control SD, $n = 5$ for control TB, $n = 12$ for iTfamKO SD, $n = 9$ for iTfamKO TB, ****$P \leq 0.0001$ for control SD versus iTfamKO SD, control TB versus iTfamKO SD, and iTfamKO SD versus iTfamKO TB, *$P = 0.0262$ for control TB versus iTfamKO TB). **H** Representative Coomassie-stained gel and quantification of urine samples ($n = 10$ for control SD, $n = 9$ for iTfamKO SD, $n = 8$ for iTfamKO TB, ****$P \leq 0.0001$ for control SD versus iTfamKO SD and control SD versus iTfamKO TB, **$P = 0.005$ for iTfamKO SD versus iTfamKO TB). **I** Kaplan–Meier survival curves ($n = 10$ for control SD, $n = 5$ for control TB, $n = 9$ for iTfamKO SD, $n = 8$ for iTfamKO TB, **$P = 0.0046$ for iTfamKO SD versus iTfamKO TB). Data are shown as means ± SEM, where each dot is a biological sample. *P*-values are determined by **B** and **E** Kruskal–Wallis test with Dunn's multiple comparisons test, **C** mixed-effects analysis with Šidák's multiple comparisons test, **D**, **G**, and **H** one-way analysis of variance (ANOVA) with Tukey's multiple comparisons test, or **I** log-rank (Mantel–Cox) test. **F** *P*-values were determined by (curve) two-way ANOVA with Tukey's multiple comparisons test or (AUC) one-way ANOVA with Tukey's multiple comparisons test. Source data and *P*-values for non-significant data are provided as a Source Data file.

peristalsis and intestinal emptying, resulting in signs of constipation as observed in patients living with mitochondrial diseases[21].

We observed similar compositional and metabolic alterations in the intestinal microbiota of iTfamKO and mtDNA-mutator mice, supporting previously reported age-associated microbiota shifts in mtDNA-mutator animals[42]. This endosymbiotic failure has also been described in mice lacking intestinal mitochondrial heat shock protein 60, which display reduced microbial diversity and expansion of disease-associated bacteria leading to intestinal pathology[35]. Altogether, these studies highlight a central role for host mitochondria in

shaping gut microbiota composition and function, either promoting homeostasis or contributing to pathological remodeling[6,43,44].

Both iTfamKO and mtDNA-mutator mice share a reduction in core SCFA-producing bacteria, suggesting conserved host–microbiota interactions linked to mitochondrial fitness. Notably, among SCFAs, only butyrate levels are diminished in the feces of mtDNA-mutator mice. This reduction correlates with decreased abundance of specialized butyrate-producing taxa, such as *Lachnospiraceae* and *Ruminococcaceae*, whereas bacteria specialized in producing acetate or propionate seem unaffected, potentially explaining the unchanged

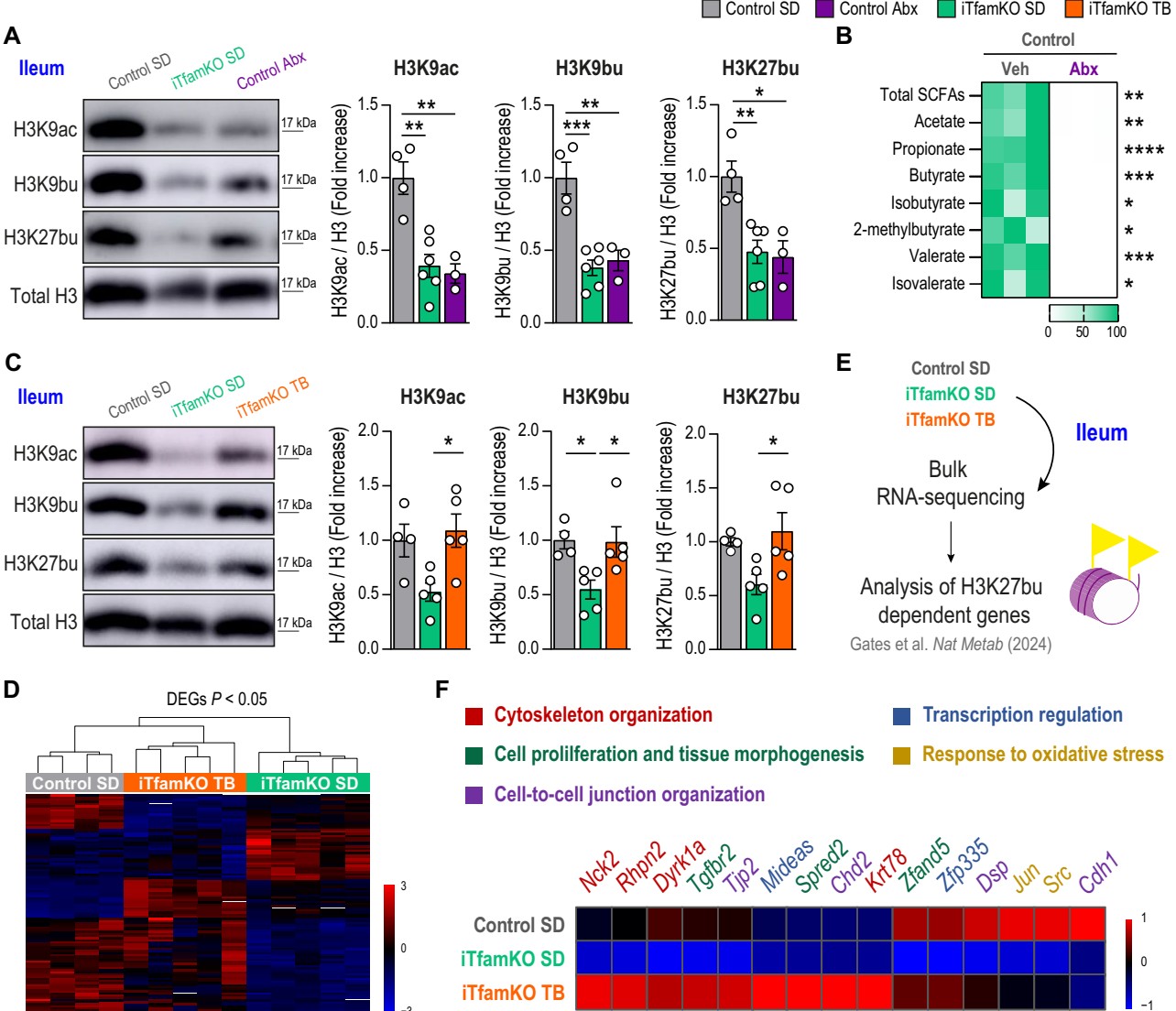

**Fig. 7 | Tributyrin administration restores microbiota-dependent epigenetic marks in the ileum of iTfamKO mice. A** Representative immunoblot and quantification of histone H3 acetylation and butyrylation marks in the ileum of control, iTfamKO, and antibiotic (Abx)-treated control mice ($n = 4$ for control SD, $n = 6$ for iTfamKO SD, $n = 3$ for control Abx; H3K9ac: **$P = 0.0016$ for control SD versus iTfamKO SD, **$P = 0.0029$ for control SD versus control Abx; H3K9bu: ***$P = 0.0004$ for control SD versus iTfamKO SD, **$P = 0.0026$ for control SD versus control Abx; H3K27bu: **$P = 0.0067$ for control SD versus iTfamKO SD, *$P = 0.0122$ for control SD versus control Abx). **B** Heatmap depicting normalized values of short-chain fatty acid (SCFA) levels in the feces of wild-type mice receiving vehicle (Veh) or Abx ($n = 3$ for control Veh, $n = 3$ for control Abx, **$P = 0.001$ for total SCFAs, **$P = 0.0023$ for acetate, ****$P \le 0.0001$ for propionate, ***$P = 0.0003$ for butyrate, *$P = 0.00159$ for isobutyrate, *$P = 0.0112$ for 2-methylbutyrate, ***$P = 0.0002$ for valerate, *$P = 0.0143$ for isovalerate). **C** Representative immunoblot and quantification of histone H3

acetylation and butyrylation marks in the ileum of control and iTfamKO mice fed either a standard diet (SD) or a 10% tributyrin-supplemented diet (TB) ($n = 4$ for control SD, $n = 5$ for iTfamKO SD, $n = 5$ for iTfamKO TB; H3K9ac: *$P = 0.0225$ for iTamKO SD versus iTfamKO TB; H3K9bu: *$P = 0.0395$ for control SD versus iTfamKO SD, *$P = 0.0353$ for iTfamKO SD versus iTfamKO TB; H3K27bu: *$P = 0.035$ for iTfamKO SD versus iTfamKO TB). **D** Hierarchical clustering of differentially expressed genes (DEGs) in bulk RNA-sequencing analysis in the ileum. **E** Experimental design of the transcriptomic analysis of genes whose expression was previously reported to be dependent on the H3K27bu mark. **F** Heatmap depicting expression of H3K27bu-dependent genes in ileum RNA-sequencing data. Data are shown as means ± SEM, where each dot is a biological sample. $P$-values were determined by **A** and **C** one-way analysis of variance (ANOVA) with Tukey's multiple comparisons test or **B** unpaired two-tailed Student's $t$-test. Source data and $P$-values for non-significant data are provided as a Source Data file.

levels of these metabolites. Because butyrate-producing bacteria depend on an intact mucus layer[33], impaired goblet cell function in iTfamKO mice may underlie the loss of SCFA-producing bacteria. These findings align with previous reports showing that mitochondrial function is required for proper mucin deposition[45] and that disruption of the mucus layer leads to the loss of beneficial SCFA-producing bacteria in both mice and humans[46–48]. On top of this, mitochondrial metabolism modulates oxygen bioavailability in the intestinal lumen[34], and changes in oxygen levels may further alter the composition of the commensal microbiota.

Beyond intestinal complications[49], FMT strategies appear to be a feasible tool to alleviate cardiovascular[50], metabolic[51], and neurological conditions[52]. In this regard, we have found that transplantation of fecal microbiota from healthy control mice delays signs of disease and extends lifespan in iTfamKO mice, which is associated with restored levels of butyrate. Notably, replenishing butyrate mitigates sarcopenia and improves glucose handling, metabolism, and kidney function in iTfamKO mice. In line with this, strategies to remodel the intestinal microbiota and its derived SCFAs showed beneficial effects in *Ndusf4*[−/−] mice that mimic Leigh syndrome, a mitochondrial disease that courses

with neurological symptoms[53]. Likewise, strategies to restore eubiosis and increase butyrate levels exerted beneficial effects in mouse models of amyotrophic lateral sclerosis[54,55], a condition closely linked to mitochondrial dysfunction[56]. Beyond enhancing health outcomes, butyrate administration can extend longevity in different experimental models[55,57], similar to that detected in iTfamKO mice after tributyrin administration. Of note, we observed that tributyrin treatment is substantially more effective than FMT in iTfamKO mice, likely reflecting distinct mechanisms of action. While tributyrin directly provides a sustained source of butyrate, thereby bypassing the need for bacterial colonization, the FMT strategy relies on the successful engraftment of a healthy microbiota into a hostile intestinal environment, thus limiting its beneficial impact in iTfamKO mice.

Mechanistically, butyrate can modulate post-translational modifications of histones through inhibition of histone deacetylases, and acting as substrate for direct acetylation, propionylation, and butyrylation, linking microbial metabolism to the epigenetic landscape of the host[24,27,58]. Particularly, we observed that the levels of butyrylated histone H3 and expression of several butyryl-H3-dependent genes that are crucial for intestinal homeostasis were decreased in the ileum of iTfamKO mice. Recovering these epigenetic marks upon TB supplementation was associated with upregulation of genes involved in the physical barrier and the oxidative stress response in the intestine of these mice. This SCFA-dependent regulation of histone acylations provides a molecular framework by which microbial cues dictate gene expression and host health outcomes. Since the therapeutic effects of tributyrin can be attributed to other molecular mechanisms, future studies should clarify the relevance of butyrate-dependent epigenetic remodeling in disorders showing mitochondrial dysfunction. For example, butyrate can also serve as an agonist for G-coupled protein receptors and transcription factors, such as Peroxisome Proliferator-Activated Receptors (PPARs)[25,34,58]. Accordingly, PPAR agonists have been shown to boost mitochondrial fitness in a myriad of pathologies presenting with a decline in mitochondrial function[59–62]. In cells with competent mitochondria, butyrate can also act as an energy source, which can be metabolized into acetyl-CoA through fatty acid oxidation to help sustain cellular bioenergetics[58]. Upon inducible TFAM depletion, the extent of mitochondrial turnover could create a therapeutic window for butyrate to act as an alternative bioenergetic substrate.

Butyrate, through its dual role in regulating metabolism and the epigenetic landscape, offers potential therapeutic benefits in various conditions related to mitochondrial dysfunction. Hence, it is tempting to speculate that strategies aiming to restore host–microbiota symbiosis or to normalize microbiota-derived metabolites could ameliorate the progression of diseases associated with deficient mitochondrial function.

## Methods

### Animal procedures and diets

All animal experimentation procedures were authorized by the Animal Experimentation Ethics Committees of Centro de Biología Molecular Severo Ochoa (CBM) and Centro Superior de Investigaciones Científicas (ProEx 20.5/25), making every effort to minimize mouse discomfort. $Tfam^{fl/fl}Ubc^{Cre-ERT2}$ mice were generated by crossing $Tfam^{fl/fl}$ mice with mice expressing the inducible Cre recombinase Cre-ERT2 under the control of the ubiquitin gene ($Ub^{Cre-ERT2}$ mice). $Tfam^{fl/fl}Ubc^{Cre-ERT2}$ mouse colony was bred and maintained in the CBM Animal Facility under specific pathogen-free conditions. Both $Tfam^{fl/fl}Ubc^{Cre-ERT2-/-}$ and $Tfam^{fl/fl}Ubc^{Cre-ERT2+/-}$ mice were intraperitoneally administered 20 mg/ml of tamoxifen (Sigma-Aldrich) dissolved in a 10% ethanol, corn oil solution for five consecutive days. For tributyrin supplementation studies, mice were fed either a standard diet (Research Diets, D11112201i) or a 10% (w:w) tributyrin-supplemented standard diet (Research Diets, D24012601i). $PolgA^{D257A/D257}$ ($Polg^{mut}$) or

mtDNA-mutator mice and their controls, $PolgA^{wt}$ mice, were bred and maintained in Centro Nacional de Investigaciones Cardiovasculares (CNIC) Animal Facility under specific pathogen-free conditions. Two to five mice were housed per cage, separated by genotype and sex, and fed ad libitum, receiving cardboard materials as part of the environmental enrichment. Mice were bred on a C57BL/6J background, and most studies were performed in female mice to facilitate microbiota normalization by mice and/or cage swapping before starting the experiment.

**Glucose tolerance tests.** After determination of overnight fasted blood glucose levels, mice were intraperitoneally injected with glucose at a dose of 2 g per kilogram of body weight [10% (w:v)]. Afterwards, blood glucose levels were determined from the blood of mouse tails at 15, 30, 60, 120, and 180 min using Contour Next reactive glucose strips and a glucometer (Bayer).

**Body temperature measurements.** Body temperature was measured in immobilized mice with their abdomen exposed. An infrared thermometer sensor was then placed below the lower abdomen at approximately 2–5 mm away from the abdomen surface while holding the mouse with its body parallel to the ground. Three measurements were performed for each mouse.

**Forelimb grip strength assessment.** Mice were held by their tails, and forelimb grip strength was measured as tension force using a digital force transducer (Grip Strength Meter, Bioseb). Five to seven measurements per trial were performed for each mouse, with a few seconds resting period between measurements.

**Rotarod test.** Motor coordination was assessed in an accelerating rotarod apparatus (Ugo Basile). Mice were trained for two consecutive days at constant speed: on the first day, four times at 4 revolutions per minute (rpm) for 1 min and, on the second day, four times at 8 rpm for 1 min. The day afterwards, the rotarod instrument was set to progressively accelerate from 4 to 40 rpm for 5 min. During the accelerating trials, the latency to fall from the rod was measured. Mice were tested four times.

**Open field test.** For behavioral experiments, mice were simultaneously transferred to a behavior room where they habituated for several days before the start of the tests. The light/dark cycle was 12/12 h (lights on at 8:00 am). All the experiments were performed during the light phase. To minimize variability, each animal was always tested at the same time of the day. The open field test was measured in a $43 \times 43$ cm square test arena and recorded for 3 days. Mice were first habituated to the dark for 30 min and then placed in the illuminated arena for 10 min. Free movement was recorded and analyzed using ANY-maze™ Video Tracking System software (Stoelting Co.). To assess locomotion, total distance traveled, exploration speed, and time immobile were quantified.

**Nest-building test.** Nest-building performance was performed as formerly described[63]. Briefly, mice were housed individually overnight in clean cages containing bedding but no environmental enrichment. Then, an intact new nestlet was placed inside the cage. Next morning, the nest building was scored as follows: nestlet mostly untouched (1 point); nestlet partially torn but mainly flat (2 points); nestlet mostly torn but no nest identifiable (3 points); identifiable but mainly flat nest (4 points); perfectly built nest with walls covering the mouse body (5 points).

**Hematological analyses.** Blood was extracted by submandibular vein puncture in Microvette® EDTA K2 tubes. Supernatant plasma was collected from anticoagulated samples after centrifugation at $10,000 \times g$

for 12 min at 4 °C. Finally, plasma was assessed with an Abacus Junior or Element HT5 hematology analyzer.

**Intestinal permeability assay.** Mice fasted 2 h were orally gavaged with a 1:3–1:4 (v:v) dilution of 250 mg/ml 4-kDa fluorescein isothiocyanate (FITC)-dextran probe (Sigma-Aldrich) at a dose of 0.6 g per kilogram of body weight. After 2 h, 100 to 120 μl of blood was collected in BD Microtainer® tubes from the facial vein of mice and centrifuged at $6000 \times g$ for 10 min at 4 °C to obtain the serum fraction. FITC-dextran measurements were performed in duplicates by fluorimetric quantification of mouse serum mixed with an equal volume of phosphate-buffered saline (PBS) [1:9 (v:v)]. Dilutions of non-treated mouse serum in FITC-dextran diluted with PBS were used as a standard curve to calculate blood FITC-dextran concentrations. One hundred microliters of standards or diluted sera was measured in a FLUOstar OPTIMA® (BMG Labtech) 96-plate reader at an excitation wavelength of 492 nm and an emission wavelength of 525 nm.

**Fecal output and water content analyses in feces.** Mice were housed individually overnight in clean cages without bedding or environmental enrichment, and the number of feces per cage was enumerated the next day. A fresh fecal sample from each animal was immediately collected after caging, weighed, and dried at 50 °C overnight. The next morning, samples were weighed again, and the difference was considered the fecal water content and expressed as a percentage.

**Fecal microbiota transplantation (FMT).** FMTs were performed as formerly described[64]. In brief, mice fasted 6 h were orally gavaged for 3 consecutive days with 200 μl of an antibiotic cocktail consisting of neomycin (1 mg/ml) (Nzytech), ampicillin (1 mg/ml) (Nzytech), metronidazole (1 mg/ml) (Sigma-Aldrich), and vancomycin (0.5 mg/ml) (Sigma-Aldrich) in autoclaved water. The day afterwards, four to five fresh fecal pellets were pooled from donor mice in 600 μl of reduced buffer (0.5 mg/ml cysteine and 0.2 mg/ml $Na_2S$ in PBS) and vortexed for 1 min. Homogenates were then centrifuged at $500 \times g$ for 5 min to remove large particles. Finally, 200 μl of fecal slurry was orally gavaged to recipient mice fasted for 4 h twice a week for 2 weeks and then, once a week until sacrifice. Following FMT, the remaining slurry was applied to the fur of recipient mice, and their cages were replenished with fresh fecal pellets and dirty bedding from donor mice to ensure coprophagia.

**Microbiota depletion experiments.** Mice were randomly assigned to vehicle or antibiotics (Abx) treatment groups. Mice in the latter were administered a cocktail of neomycin (1 mg/ml) (Nzytech), ampicillin (1 mg/ml) (Nzytech), metronidazole (1 mg/ml) (Sigma-Aldrich) and vancomycin (0.5 mg/ml) (Sigma-Aldrich) in autoclaved drinking water supplemented with sucrose (2 mg/ml) to improve palatability and renewed weekly. Vehicle mice were treated with sucrose water without antibiotics for the same time period.

## Histological, immunohistochemical, and immunofluorescence analysis

**Histological analysis.** Tissues were collected from euthanized mice after intracardiac perfusion with cold PBS. Organs were then fixed in 10% neutral buffered formalin for 24 h and dehydrated in 70% ethanol until processing. Dehydrated organs were embedded in paraffin and further processed with a microtome to obtain sections. Afterwards, sections were deparaffinized and stained as detailed in the figure legends for their histological examination. Images were captured using the 5× objective of a vertical microscope AxioImager M1 (Zeiss) connected to a DMC6200 camera (Leica) and the LasX software (v4.13). Crypt and villus lengths were measured from the bottom of the crypt to the crypt–villus junction, and from there to the tip of the villus, respectively. Quantification of Periodic Acid-Schiff (PAS) or Alcian

Blue–stained areas was performed using ImageJ software (v1.53n) following image deconvolution. The number of PAS-positive cells (goblet cells) in the ileum and colon was enumerated and normalized to villus or crypt length, respectively. Areas identified as PAS or Alcian Blue-positive by thresholding were measured and then normalized to the total tissue area. Twenty-five to fifty crypt–villus units per mouse were quantified in a single-blind manner. Elastin lamina breaks, defined as interruptions in the elastin fibers, were counted in the entire medial layer of 3 consecutive cross-sections per mouse, and the mean number of breaks was calculated. Image quantification was performed using the NanoZoomer Digital Pathology software (v2.7.25).

**Immunofluorescence analysis.** For brain immunofluorescence staining, the left hemisphere of each brain was fixed by immersion in 4% paraformaldehyde (PFA) in PBS for 24 h at room temperature (RT) and stored in PBS at 4 °C. Then, fixed hemispheres were embedded in a solution of 10% sucrose–4% agarose in phosphate buffer (PB). The resulting blocks were cut with a vibratome (Leica), obtaining 40-μm-thick sagittal sections, which were stored at −20 °C in 40% glycerol, 30% ethylene glycol in 0.2 M PB, pH 7.4. Brain slices were washed with Tris-buffered saline (TBS), permeabilized, and blocked with blocking solution (0.5% Triton X-100 and 2% BSA in TBS) for 1 h at RT. Sections were then incubated overnight with mouse anti-NeuN (1:500 dilution; Millipore, MAB377) diluted in blocking solution at 4 °C. After 3 washes with TBS, each section was incubated with secondary antibodies conjugated to fluorophores for 1 h at RT. They were washed again with TBS and nuclei were counterstained with DAPI for 5 min (1/5000 dilution; Merck, 268298). Finally, sections were mounted on slides using Prolong Glass Mounting Medium (Invitrogen). Images were acquired using the oil immersion 25× objective of an LSM710 confocal microscope coupled with a vertical microscope AxioImager M2 (Zeiss) and the ZEN Black 2010 software and were analyzed with the ImageJ software (v1.53n). Lipofuscin granules were quantified in a single-blind manner in brain slides owing to their autofluorescent properties.

**Electron microscopy.** After collection, a portion of the ileum and the colon were fixed with 4% PFA 2% glutaraldehyde in 0.1 M PB, pH 7.4. Tissues were post-fixed with 1% osmiun tetroxide and 1% potassium ferrocyanide in water for 1 h at 4 °C, dehydrated with lowering concentrations of ethanol, and finally embedded in resin EPON. Eighty nm sections of the embedded tissue were obtained using an Ultracult E ultramicrotome and mounted on carbon-coated copper slot grids. Sections were counterstained with 2% uranyl acetate and lead citrate for 1 h. Preparations were examined with a transmission electron microscope (JEM1400 Flash, Jeol), and images were acquired with a CMOS Oneview camera (Gatan). The area of mitochondria and the number of body inclusions per mitochondria were quantified in a single-blind manner using the ImageJ software (v1.53n).

**Immunohistochemical analysis.** For immunohistochemical staining, deparaffinized ileal sections were rehydrated and boiled in citrate buffer (10 mM citrate buffer, 0.05%Triton X-100, pH 6) to retrieve antigens. Then, sections were blocked in 10% goat serum, 5% horse serum, 0.05% Triton X-100, and 2% BSA in PBS for 45 min. Endogenous peroxidase and biotin were blocked with 1% hydrogen peroxide-methanol for 10 min and a biotin blocking kit (Vector Laboratories), respectively. Sections were incubated with a goat anti-Ki67 antibody (RD Systems, AF3667) and color was obtained with 3,3′-diamino-benzidine (Vector Laboratories). Sections were counterstained with hematoxylin and mounted in DPX (Fluka). Slides were scanned with a NanoZoomer-RS scanner (Hamamatsu). The number of Ki67-positive cells was quantified in a single-blind manner using NanoZoomer Digital Pathology software (v2.7.25).

## Immunoblot

For Western blot analysis, frozen tissues were dissected and homogenized in RIPA buffer (20 mM Tris−HCl, pH 7.5, 150 mM NaCl, 1 mM EDTA, 1 mM EGTA, 1% NP−40, 1% sodium deoxycholate, 0.1% SDS) with proteases (cOmplete™, Sigma-Aldrich) and phosphatases inhibitors (Sigma-Aldrich). The concentration of proteins in homogenized samples was determined using the Pierce™ BCA Protein Assay Kit (Thermo Fisher Scientific) or the DC Protein Assay (Bio-Rad), following the manufacturer's instructions. Thirty to 50 μg of brain lysates, or 20 μg of ileum lysates, were prepared in Laemmli buffer (25 mM Tris−HCl pH 6.8, 1% SDS, 3.5% glycerol, 0.4% 2-mercaptoethanol, and 0.04% bromophenol blue) and separated by electrophoresis in polyacrylamide gels in the presence of sodium dodecylsulfate (SDS) at constant voltage.

Proteins were then transferred to nitrocellulose membranes by wet transfer for brain lysates or using the Trans-Blot® Turbo™ system (Bio-Rad) for ileum lysates and, after blocking with 5% BSA or 5% milk in 0.1% Tween-20 in TBS [T-TBS], membranes were incubated overnight with rabbit anti-TFAM (1:1000 dilution; Proteintech, 22586), rabbit anti-calnexin (1:10,000 dilution; Abcam, ab22595), rabbit anti-tau-P Ser 396 (1:1000 dilution; Life Technologies, 44752G), mouse anti-PSD-95 (1:1000 dilution; BD Biosciences, 610495) or mouse anti-vinculin (1:10,000 dilution; Abcam, ab129002) for brain lysates, or rabbit anti-H3K9ac (1:5000 dilution; Abcam, ab4441), rabbit anti-H3K9bu (1:500 dilution; PTMBio, PTM-305), rabbit anti-H3K27bu (1:1000 dilution; Merck, ABE2854), or rabbit anti-H3 (1:1000 dilution; Abcam, ab1791) for ileal lysates diluted in T-TBS at 4 °C with gentle agitation. After washing with T-TBS, membranes were incubated with the corresponding secondary antibodies coupled to horseradish peroxidase at 1:2500 to 1:15,000 dilution for 1 h at RT. Protein signal was detected with Pierce™ ECL Western Blotting Substrate (Thermo Fisher Scientific), and chemiluminescence was measured with an Amersham™ Imager 680 (GE HealthCare). Each protein of interest was quantified using the ImageJ software (v1.53n) by measuring the average pixel intensity of the corresponding band and normalizing the value to the respective control protein. Full uncropped scans of any cropped blot images are provided in the Supplementary Information and Supplementary Data files.

## RNA Isolation and qPCR Analysis

Total RNA extraction was performed with a MagNA Lyser® homogenizer (Roche) using 1 cm of frozen tissue in 700 μl of TRIzol® reagent (Invitrogen). The RNA from the resulting aqueous phase was further purified using the RNeasy Mini Kit (QIAGEN) following the manufacturer's instructions. Quality and quantity were determined by measuring the absorbance at 260 and 280 nm on a Nanodrop One spectrophotometer (Thermo Fisher Scientific).

cDNA libraries were prepared from total RNA and then validated and quantified by an Agilent 4200 TapeStation in Haplox. After passing library inspection, stranded mRNA was sequenced either on the Illumina Novaseq Xplus platform, and FastQ files were generated containing nucleotide data and quality scores for each position. The quality of FastQ files was checked using FastQC (v0.11.9). RNA-sequencing reads were mapped to the *Mus musculus* reference genome GRCm39 using either Hisat2 (v2.2.1) or STAR (v2.5.2) software. Reads were then pre-processed with SAMtools (v1.13) to transform Sequence Alignment/Map files into Binary Alignment/Map files and sorted. The number of reads covered by each gene was calculated by HTSeq-Count (v1.99.2).

Downstream data analysis was performed with R (v4.3.2). Differential gene expression (DEG) analysis was performed using DESeq2 (v1.44). Genes with $P$-value < 0.05 and |log$_2$-fold change| > 0.6 were determined to show statistically significant differences in group comparison. Over-representation analysis (ORA) and gene set enrichment analysis (GSEA) were performed using the clusterProfiler (v4.8.3)

package in GO, KEGG, WikiPathways, Reactome, and the Hallmarks of the Molecular signatures databases. PCA plots, chord diagrams, and heatmaps were visualized by using ggplot2 (v3.4.4), circlize (v0.4.15), and pheatmap (v1.0.12), respectively.

For qPCR analysis, reverse transcription was performed using 500 ng of RNA extracts and the Maxima™ First Strand cDNA Synthesis Kit and dsDNase (Thermo Fisher Scientific). The reaction was performed in a ProFlex™ PCR System thermocycler (Applied Biosystems) at 25 °C for 10 min, followed by 50 °C for 15 min, and 85 °C for 5 min. cDNA samples were then cooled to 4 °C and stored at −20 °C for qPCR analysis. Amplification conditions were determined by the primers to present amplification efficiency close to 100% and a single peak in melt-curve analyses. We included 1:15-1:25 dilution of cDNA samples in the qPCR reaction, along with 5 μl of polymerase (Go Taq® Master Mix, Promega) and 0.5 μl of each forward and reverse primer solution (5 μM stock) added to 384-well plates. The reaction was run on a Bio-Rad CFX 384 thermocycler. The 5′−3′ primer pairs (FW, RV. Sigma) used for qPCR are listed in Table S1.

Quantitative RT-PCR data were analyzed using the $2^{-\Delta\Delta Ct}$ method to calculate the relative changes in gene expression, where ΔΔCt is the difference between the problem sample Ct values and the control ones. The relative expression in each figure represents the induction levels of the gene of interest relative to *Hprt*, *Pp1a*, or *B2m*.

## Urine protein analysis

Urine was collected from mice after immobilizing them on top of a piece of Parafilm and immediately frozen at −80 °C. Ten microlitres of urine samples or 0.05 mg/ml BSA (Bio-Rad) in Laemmli buffer were loaded into a 12% polyacrylamide gel in the presence of SDS at constant voltage to perform electrophoresis. Electrophoresis gel was then stained with a 0.1% Coomassie blue (w:v) in 20% methanol and 10% acetic acid solution by heating the gel for 1 min in a microwave and placing it on a shaker for 30 min at RT. Afterwards, gel was destained using a 20% methanol 10% acetic acid solution by heating it for 30 s in a microwave and then washing it several times with the indicated solution until bands start to be visible. Images were acquired with an Amersham™ Imager 680 (GE HealthCare). The band corresponding to albumin was quantified using the ImageJ software (v1.53n). Full uncropped scans of any cropped gel images are provided in the Supplementary Data file.

## Luminex detection of cytokines

Blood (100–120 μl) was collected in BD Microtainer® tubes from the facial vein or after culling mice and centrifuged at 6000 × *g* for 10 min at 4 °C to obtain the serum fraction. Cytokines from serum samples were quantified using the multiplexed bead-based immunoassay Cytokine & Chemokine 26-Plex Mouse ProcartaPlex™ Panel 1 (Invitrogen, EPX260-26088-901) following the manufacturer's instructions.

## LPS-binding protein (LBP) ELISA quantification

Serum samples from mice were diluted and processed following the manufacturer's instructions (Enzyme Immunoassay for quantification of mouse LBP, Biometec). The plate was measured in a Dynex Opsys MR™ 96-plate reader (Aspect Scientific) at an excitation wavelength of 450 nm for absorbance quantification, and at 630 nm for wavelength correction.

## Microbiota analysis by *16S* rRNA gene sequencing

*16S* rRNA gene sequence analyses were performed as formerly reported[65]. In brief, bacterial DNA extracted from samples collected from either total ileal and colon contents, or expelled feces was extracted using the E.Z.N.A stool DNA kit (Omega Biotek) following the manufacturer's instructions. Amplicons of the V4 region of the *16S* rRNA gene were obtained in each sample using the following 5′−3′ primer pair: CCTACGGGAGGCAGCAG (FW) and ATTACCGCGGCTGCTGG (RV).

Libraries were then sequenced using an Illumina MiSeq instrument, and sequences were curated and analyzed using the mothur (v.1.40.5) software package.

## Short-chain fatty acid (SCFA) quantification by UPLC-MS/MS

Standards of eight straight and branched-chain SCFAs [acetic acid (AA), propionic acid (PA), isobutyric acid (i-BA), butyric acid (BA), 2-methylbutyric acid (2-Me-BA), isovaleric acid (i-VA), valeric acid (VA), and 3-methylvaleric acid (3-Me-VA)], 3-nitrophenylhydrazine (3NPH), N-(3-dimethylaminopropyl)-N'-ethylcarbodiimide hydrochloride (EDC), formic acid, and pyridine were purchased from Sigma-Aldrich. Acetonitrile (ACN) was purchased from VWR. Calibration curves were constructed using a mixed SCFA solution in ACN-water (50:50, v/v), with ranges of 1–10,000 μM for AA and 0.1–1000 μM for PA, i-BA, BA, 2-Me-BA, i-VA, and VA.

SCFAs were extracted from 50 mg of fecal, ileal, cecal, or colonic contents using 100 μl of ACN-water (50:50, v/v) containing 5 μM internal standard, homogenized using a FastPrep-24 5 G system (MP Biomedicals), incubated at 800 rpm for 15 min at 10 °C, and centrifuged at $14,000 \times g$ for 30 min at 4 °C. For serum samples, SCFAs were extracted by adding 60 μl of ice-cold methanol (containing IS) to 30 μl of serum, following incubation on ice for 1 h. Then, the suspension was centrifuged at $14,000 \times g$ for 15 min at 4 °C. Forty microliters of the supernatant were then mixed with 20 μl of 200 mM 3-nitrophenylhydrazine (3NPH) and 120 mM EDC/6% pyridine in ACN-water (50:50, v/v), followed by incubation for 30 min at 40 °C. After cooling to room temperature, fecal, cecal, and colonic samples were diluted 20-fold with 10% ACN in water, whereas serum and ileal content samples were diluted 2-fold or 4-fold, respectively, with 10% ACN in water. All samples were subsequently centrifuged at $14,000 \times g$ for 10 min at 4 °C and analyzed by UPLC-MS/MS. Standards and blanks were processed using the same derivatization protocol.

SCFA analysis was conducted following a previously described protocol[66]. Analyses were carried out using an Agilent 1260 Infinity II system coupled to an Ultivo 6465 Triple Quadrupole LC-MS, equipped with an Agilent Jet Stream ESI source and controlled via MassHunter Workstation (Agilent Technologies). Multiple reaction monitoring (MRM) acquisition was conducted using the optimal transitions and parameters listed in Table S2.

## Statistical analyses, reproducibility, and figure design

Statistical analyses were performed using GraphPad Prism 9 or Past 3.22 software. No statistical method was used to predetermine sample size. Outliers were identified and excluded by the ROUT method (5%). If data followed a normal distribution after applying the Shapiro−Wilk test, comparisons between two datasets were performed using the unpaired two-tailed Student's $t$-test, and comparisons between more than two datasets were performed using the one or two-way analysis of variance (ANOVA) or mixed-effects analysis with Tukey's or Šidák's multiple comparison tests. If not, comparisons between two or more datasets were calculated using the non-parametric two-tailed Mann−Whitney $U$ and Wilcoxon tests or Kruskal−Wallis $H$ test with Dunn's multiple comparison test, respectively. For survival curves, the log-rank Mantel−Cox test was applied. Permutational multivariate analysis of variance (PERMANOVA) was performed based on 9999 permutations. Differences with $P$-values $\leq 0.05$ were considered significant. *$P \leq 0.05$; **$P \leq 0.01$; ***$P \leq 0.001$; and ****$P \leq 0.0001$. Unless otherwise stated, experimental data were represented as means ± the standard error of the mean (SEM), where each dot was an individual biological sample of each experimental group. Box-and-whisker plots represent the interquartile range between the first and third quartiles (25th and 75th percentiles, respectively), the median, and the maximal and minimal values. In the figure legends, $n$ denotes the number of pooled mice per group in the experiment. Figures were designed using GraphPad Prism 9 and Adobe Illustrator (v29.0.1).

## Reporting summary

Further information on research design is available in the Nature Portfolio Reporting Summary linked to this article.

## Data availability

Ileum RNA-sequencing raw data, as well as microbiota *16S* rRNA gene sequencing raw data, are publicly available via NCBI with BioProject numbers PRJNA1301342 and PRJNA1291203, respectively. Source data are provided with this paper.

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

## Acknowledgements

We sincerely thank R. Caruso, M. Hasewaga, and the Microbiome Core of the University of Michigan, as well as the Electron Microscopy and the Advanced Optical Microscopy (SMOA) facilities of Centro de Biología Molecular Severo Ochoa (CBM). We also thank the Oxford Hospitals charity and Lily Foundation "Treatments for Mitochondrial Disease – 2019/20" (J.P.), Fundación Ramón Areces and MINECO through the Centers of Excellence Severo Ochoa Award, and CERCA Programme of the Generalitat de Catalunya (A.V.L.-V.). Funded by the European Union (M.M.). Views and opinions expressed are, however, those of the authors only and do not necessarily reflect those of the European Union or the European Research Council Executive Agency. Neither the European Union nor the granting authority can be held responsible for them. This research was supported by a European Research Council grant ERC-2021-CoG 101044248-Let T Be (M.M.), Comunidad de Madrid (Spain) grant Y2020/BIO-6350 NutriSION-CM synergy (E.C. and M.M.), Spanish Ministerio de Ciencia e Innovación grant PID2022-141169OB-I00 (M.M.), Comunidad de Madrid (Spain) – Universidad Autónoma de Madrid (UAM) grant SI4/PJI/2024-00166 (E.C.), NIH grant R01 DK095782 (G.N.), Ministerio de Ciencia, Innovación y Universidades (Spain) FPU grants FPU19/02576 (M.M.G.H.), FPU24/02492 (P.R.-R.) and FPU20/04066 (J.I.E.-L.), Comunidad de Madrid (Spain) PIPF grant PIPF-2022/SAL-GL-25208 (S.D.-P.), Universidad Autónoma de Madrid FPI-UAM grant (G.S.-H.), and Ministerio de Ciencia, Innovación y Universidades (Spain) Juan de la Cierva-Incorporación grants IJC2018-036850-I (E.G.-R.) and JC2020-044392-I (I.F.-Q.).

## Author contributions

Conceptualization: E.G.-R., M.M.G.H, and M.M. Formal analysis: E.G.-R., M.M.G.H., P.R.-R., C.S., V.G.-C., N.I., I.B.-L., V.E.-Z., A.F-A., J.O., G.S.-H., E.C., S.D.-P., and J.F.A. Funding acquisition: E.C., M.M., and G.N. Investigation: E.G.-R., M.M.G.H., P.R.-R., C.S., V.G.-C., N.I., I.B.-L., V.E.-Z., J.O., A.F.-A., G.S.-H., E.C., C.V.-M., S.D.-P., J.I.E-L., I.F.-Q., and J.F.A. Resources: R.J.-M., J.P., A.V.L.-V., J.A.E., and G.N. Visualization: E.G.-R., M.M.G.H, and M.M. Writing—original draft: E.G.-R., M.M.G.H, and M.M. Writing—review and editing: E.G.-R., M.M.G.H, and M.M.

## Competing interests

The authors declare no competing interests.

## Additional information

Enrique Gabandé-Rodríguez ®[1,17], Manuel M. Gómez de las Heras ®[1,2,17], Pablo Ramírez-Ruiz de Erenchun ®[1,2], Carolina Simó[3], Virginia García-Cañas ®[3], Naohiro Inohara[4], Inés Berenguer-López[5], Violeta Enríquez-Zarralanga[5], Álvaro Fernández-Almeida ®[1], Jorge Oller[6,7], Gonzalo Soto-Heredero ®[1], Elisa Carrasco ®[1,8,9,10], Cristina Vázquez-Muñoz[8,10], Sandra Delgado-Pulido[1,8], José Ignacio Escrig-Larena ®[1,2], Isaac Francos-Quijorna[1], Raquel Justo-Méndez ®[11], Juan Francisco Aranda[12], Joanna Poulton[13], Ana Victoria Lechuga-Vieco ®[14], José Antonio Enríquez ®[11,15], Gabriel Núñez[4,16] & María Mittelbrunn ®[1] ✉

[1]Tissue and Organ Homeostasis Program, Centro de Biología Molecular Severo Ochoa (CBM), Consejo Superior de Investigaciones Científicas (CSIC) - Universidad Autónoma de Madrid (UAM), Madrid, Spain. [2]Departament of Molecular Biology, Faculty of Science, Universidad Autónoma de Madrid (UAM), Madrid, Spain. [3]Molecular Nutrition and Metabolism, Institute of Food Science Research (CIAL), Consejo Superior de Investigaciones Científicas (CSIC) - Universidad Autónoma de Madrid (UAM), Madrid, Spain. [4]Department of Pathology and Rogel Cancer Center, University of Michigan Medical School, Ann Arbor, MI, USA. [5]Physiological and Pathological Processes Program, Centro de Biología Molecular Severo Ochoa (CBM), Consejo Superior de Investigaciones Científicas (CSIC) - Universidad Autónoma de Madrid (UAM), Madrid, Spain. [6]Laboratory of Vascular Pathology, IIS-Fundación Jiménez Díaz, Madrid, Spain. [7]Centro de Investigación Biomédica en Red de Enfermedades Cardiovasculares (CIBERCV), Faculty of Medicine and Biomedicine, Universidad Alfonso X El Sabio, Madrid, Spain. [8]Department of Biology, Faculty of Sciences, Universidad Autónoma de Madrid (UAM), Madrid, Spain. [9]Instituto Universitario de Biología

Molecular-IUBM, Universidad Autónoma de Madrid (UAM), Madrid, Spain. [10]Instituto Ramón y Cajal de Investigación Sanitaria (IRYCIS), Madrid, Spain. [11]Cardiovascular Regeneration Program, Centro Nacional de Investigaciones Cardiovasculares (CNIC), Madrid, Spain. [12]Department of Genetics, Physiology and Microbiology, Faculty of Biological Sciences, Complutense University of Madrid (UCM), Madrid, Spain. [13]Nuffield Department of Women's and Reproductive Health, University of Oxford, Oxford, UK. [14]Institute for Research in Biomedicine (IRB Barcelona), The Barcelona Institute of Science and Technology, Barcelona, Catalonia, Spain. [15]Centro de Investigación Biomédica en Red de Fragilidad y Envejecimiento Saludable (CIBERFES), Madrid, Spain. [16]Center for Infectious Disease Education and Research (CiDER); Osaka University, Suita, Osaka, Japan. [17]These authors contributed equally: Enrique Gabandé-Rodríguez, Manuel M. Gómez de las Heras. ✉e-mail: mmittelbrunn@cbm.csic.es

