## [Transparent Peer Review file · Nature Communications]

Butyrate extends health and lifespan in mice with mitochondrial deficiency

Corresponding Author: Dr Maria Mittelbrunn

Version 0:

Reviewer comments:

Reviewer #1

(Remarks to the Author)

The lab of Dr. Mittelbrunn and others describe phenotypes of their inducible TFAM knockout (iTfamKO) mice, which is a model of mitochondrial disease. This mouse has substantially reduced lifespan and pathologies across many different systems. The authors particularly focus on the increased intestinal barrier permeability in both the iTfamKO and another model of mitochondrial dysfunction. They demonstrate altered gut microbiota, particularly in the distal small intestine, and that treatment with fecal transplantation or tributyrin can partially rescue effects of the iTfamKO. In addition, they propose one mechanism of butyrate action could be through modulating histone modifications and gene expression. Overall, this study is compelling, comprehensive, and written in a clear manner. However, why the authors focused on the intestine versus other pathologies is not very well-defined, especially in the setting of systemic effects. In addition, some remaining questions particularly about characterizing different aspects of intestinal function, the microbiota, and SCFAs should be addressed before publication.

Major comments:

1. The authors use an inducible whole body iTfamKO model, which has wide-ranging systemic effects. How do they separate intestine-specific effects vs potential secondary effects from mitochondrial dysfunction in different tissues? The authors could use an intestine tissue-specific model to separate these effects.
2. It is unclear why the small intestine is affected by iTfamKO but not the colon. In Figure 2B and 2C, why are the mitochondrial size and inclusion bodies more significantly altered in colon compared to small intestine? In addition, histological evaluation was performed only on the small intestine (Fig. 2E–I), but not on the colon. Since the colon is a major site where SCFAs feed the energy pathways of colonocytes, including histological evaluation of the colon would further strengthen the findings of the paper.
3. The authors focus on the ileum for their analysis of the microbiota in addition to fecal pellets. Since microbial load in the small intestine is much lower than the cecum or colon, this raises several questions. First, is there bacterial overgrowth in the small intestine with iTfamKO? In addition, the authors need to describe how microbial collection of the ileum was performed (such as mucosal scraping, collection of ileal contents, etc.).
4. The study completely ignored the fact that cecum is the major site for microbial fermentation in the mice and did not provide any analysis on cecum or cecal contents. In addition, why were SCFAs not quantified from ileal, cecal, and colonic contents in addition to feces and serum? In particular, what is the effective concentration of butyrate from tributyrin-fed mice in the ileum? This analysis would provide insights into how SCFA composition varies across different intestinal sites in different mouse models, as well as after tributyrin administration.

Minor comments:

1. In Figure 1, please clarify the age of the mice. Are all data panels from 90 days post tamoxifen injection like in 1B?
2. In Figures 1 and 2, the authors showed data for small intestine and from Figure 3 on they begin referring ileum. This creates confusion. Are all the data from ileum? If so, maintaining consistent terminology throughout would help readers follow the results more easily.

3. In Figures 2L and S3, it is unclear exactly what the heatmap and legend are depicting.
4. Fecal output was measured by placing individual mouse in an empty cage for 30 minutes and quantified the number of feces defecated in 30 min. However, this approach might cause stress to the mouse, which can affect gastrointestinal motility. A more reliable approach would be placing individual mouse in metabolic cage and collect feces over a longer duration (overnight), which would provide a more convincing comparison for the defecation output.
5. While characterizing intestinal barrier disruption in iTfamKO mice, authors left out the fact that goblet cells and the mucin they produce are crucial for intestinal epithelial barrier function. A protective mucus layer formed by the secreted mucin physically separates the intestinal epithelium from the lumen, preventing the invasion of pathogenic microorganisms. Showing mRNA/protein expression or histological staining for mucins would provide additional evidence of barrier disruption.
6. In Figure 3, the data show that 16s rRNA seq was done on ileum and colon-resident microbiota of control and iTfamKO. In contrast, in Figure 4, microbial analysis of Polgmt mice was carried out using fecal samples. Could the authors clarify the rationale for such sampling differences.
7. In Figure 5F, the study shows that FMT restores butyrate levels and to a lesser extent propionate, but has no effect on acetate in the feces of iTfamKO mice. Could the authors explain why only butyrate levels were restored, but not acetate and propionate? Also, in Figure 4F, it is surprising that only butyrate is decreased. The authors should discuss these findings.
8. The study shows that both control FMT and tributyrin supplementation ameliorate signs of multiorgan morbidity. However, the epigenetic landscape was analyzed only in the context of tributyrin treatment, not following FMT. What was the rationale for this selective approach, given that control microbiota could also potentially restore the altered histone modifications observed in the intestines of iTfamKO mice? Additionally, could the authors clarify whether they consider tributyrin or FMT to be more effective?
9. The authors should clarify that some articles they cite already link butyrate to increased lifespan (PMID: 28129947) and cite additional articles such as PMID: 25127866.

Reviewer #2

(Remarks to the Author)

Reviewer #3

(Remarks to the Author)

The manuscript "Butyrate Extends Health and Lifespan in Mice with Mitochondrial Deficiency" explores the complex interplay between mitochondrial dysfunction, gut microbiota disruption, and systemic disease. The study is original, well-designed, and carefully executed, with findings that carry significant implications for human health. The manuscript is well written and effectively highlights the broad impact of mitochondrial malfunction on overall physiological homeostasis.

Using two independent models, the authors establish clear associations between mitochondrial failure, microbiota-derived metabolite imbalance, and host health, with a particular focus on gut integrity. Moreover, the study proposes promising therapeutic perspectives for microbial metabolites — notably butyrate and its precursors — as potential clinical tools for patients with mitochondrial disorders.

Overall, this is a valuable and interesting contribution, although several points merit further clarification.

Major Comments

Causality between mitochondrial dysfunction and microbiota disruption

The main conceptual limitation concerns how causality is addressed. At times, the text implies that mitochondrial disease drives microbial dysbiosis. However, detailed examination of the data suggests that the microbiota may itself contribute causally to some of the observed disease outcomes. The therapeutic effects of butyrate further support a primary or parallel contribution of microbiota-driven processes to the phenotype.

Although disentangling these bidirectional interactions is challenging, the data do not convincingly support the notion that microbiota changes are secondary to gut barrier disruption, as proposed. The discussion already hints that microbial alterations may underlie both barrier dysfunction and intestinal pathology; I encourage the authors to expand on this interpretation.

Common microbiota signatures

Both models show consistent taxonomic shifts — notably reductions in Lachnospiraceae and Ruminococcaceae in mitochondrial-deficient animals — suggesting a shared "selective pressure" on the microbiota when mitochondrial function

is compromised. This point deserves emphasis as it may reveal conserved host–microbiota interactions linked to mitochondrial health.

Additional experimental clarification

Please indicate whether microbiota composition was analyzed at the endpoint of the FMT experiments. How does mitochondrial deficiency influence microbial profiles when control feces are transplanted into Tfam-KO mice?

To further delineate causality, a reciprocal FMT experiment transferring Tfam-KO microbiota into WT mice could provide valuable insight. Although the current design is elegant and convincing, the strong mitochondrial phenotype may mask microbiota-derived contributions to disease outcomes.

Co-housing experiments could offer a more physiological means to test transmissibility, avoiding potential confounding from antibiotic pretreatment. Even if not performed, please comment on whether such an approach was considered.

Minor Comments

In Tfam-KO mice, several SCFA reductions (acetate and propionate in serum; isovalerate in feces) are not statistically significant, yet the text does not distinguish between significant and non-significant changes. Please clarify.

Previous work has linked mitochondrial dysfunction to microbiota alterations and secondary SCFA shifts (e.g., increased mitochondrial activity in MCJ-KO mutants). It would strengthen the discussion to acknowledge some of these studies:

Houghton D, et al. *J Gerontol A Biol Sci Med Sci*. 2018; 73(5):571-578.

Juárez-Fernández M, et al. *Hepatology*. 2023; 77(5):1654-1669.

Peña-Cearra A, et al. *Gut Microbes*. 2023; 15(2):2266626.

Version 1:

Reviewer comments:

Reviewer #1

(Remarks to the Author)

The authors have addressed all of my comments in a thoughtful and comprehensive manner. I have no additional concerns or suggestions, and I recommend this paper for publication.

Reviewer #2

(Remarks to the Author)

Reviewer #3

(Remarks to the Author)

I would like to thank the authors for their efforts in addressing all the submitted questions. The main points raised in the review have been adequately addressed and clearly clarified. Therefore, I recommend the publication of the article.

Reviewer #1 (Remarks to the Author):

The lab of Dr. Mittelbrunn and others describe phenotypes of their inducible TFAM knockout (iTfamKO) mice, which is a model of mitochondrial disease. This mouse has substantially reduced lifespan and pathologies across many different systems. The authors particularly focus on the increased intestinal barrier permeability in both the iTfamKO and another model of mitochondrial dysfunction. They demonstrate altered gut microbiota, particularly in the distal small intestine, and that treatment with fecal transplantation or tributyrin can partially rescue effects of the iTfamKO. In addition, they propose one mechanism of butyrate action could be through modulating histone modifications and gene expression. Overall, this study is compelling, comprehensive, and written in a clear manner. However, why the authors focused on the intestine versus other pathologies is not very well-defined, especially in the setting of systemic effects. In addition, some remaining questions particularly about characterizing different aspects of intestinal function, the microbiota, and SCFAs should be addressed before publication.

We do appreciate the constructive and positive comments of Reviewer 1 on our manuscript.

Major comments:

1. The authors use an inducible whole body iTfamKO model, which has wide-ranging systemic effects. How do they separate intestine-specific effects vs potential secondary effects from mitochondrial dysfunction in different tissues? The authors could use an intestine tissue-specific model to separate these effects.

The main objective of this study is to generate a new mouse model that comprised the multiorgan affectations shown by mitochondrial disease patients, which frequently correlate with intestinal dysfunction (PMID: 28286566), with the final aim of addressing whether the modulation of these intestinal affectations could ameliorate systemic outcomes. Our findings suggest that disrupted intestinal function likely modulates, but not causes, the complex multiorgan phenotype observed in iTfamKO mice.

In light of previous work studying an intestine-specific TFAM deficient model using *Tfam*^{fl/fl}*Villin*^{Cre-ERT2} reporting that death (human endpoint) occurs only 10-days after tamoxifen administration (PMID: 29684311), before replicating the multiorgan affectation of iTfamKO mice (PMID: 29684311), we have discussed this point with the editor and agreed to not perform this experiment in an attempt to maintain the main focus of the manuscript.

2. It is unclear why the small intestine is affected by iTfamKO but not the colon. In Figure 2B and 2C, why are the mitochondrial size and inclusion bodies more significantly altered in colon compared to small intestine?

In addition, histological evaluation was performed only on the small intestine (Fig. 2E–D), but not on the colon. Since the colon is a major site where SCFAs feed the energy pathways of colonocytes, including histological evaluation of the colon would further strengthen the findings of the paper.

Our data show that both the ileum and the colon are affected in iTfamKO mice. Despite absence of gross alterations in colon size (Fig. 2D), we observed a comparable efficiency of *Tfam* deletion in both tissues (Fig. 2A), resulting in the formation of inclusion bodies (Fig. 2B, C), and leading to alterations in the resident microbiota (Fig. 3D, E).

Following Reviewer 1's suggestion, we further explored whether *Tfam* deletion affects the colonic architecture by histological evaluation. Haematoxylin–eosin staining revealed a significant reduction in colonic crypt size in iTfamKO mice compared to controls (New Fig. S3A), similar but to a lesser extent to that observed in the ileum (Fig. 2E, F). Complementing these data, in a new set of experiments, we have found decreased concentrations of SCFAs in the colon as well as defects in the phenotype of colonic goblet cells that suggest impaired secretory function leading to a thinner mucus layer. Altogether, these data indicate that, while there may be subtle differences between the ileum and the colon, the latter also shows alterations induced by *Tfam* deficiency.

3. The authors focus on the ileum for their analysis of the microbiota in addition to faecal pellets. Since microbial load in the small intestine is much lower than the cecum or colon, this raises several questions. First, is there bacterial overgrowth in the small intestine with iTfamKO? In addition, the authors need to describe how microbial collection of the ileum was performed (such as mucosal scraping, collection of ileal contents, etc.).

Our microbiota analyses were performed in contents collected at endpoint from the ileum and the colon of control and iTfamKO mice. We have now clarified this in Materials and Methods.

Following Reviewer 1's comment, we have now quantified the bacterial load in the ileum and in the colon of iTfamKO mice by qPCR of the *16S* rRNA gene, as previously described (PMID: 15385508). As shown in **Figure 1 for Reviewers**, while the bacterial load remains unchanged in the colon microbiota, it significantly decreases in the ileal microbiota of iTfamKO mice compared to controls. These data support that bacterial overgrowth does not occur in the small intestine of iTfamKO mice.

Figure 1 for Reviewers. Analysis of bacterial load in the intestines of iTfamKO mice. Quantification of *16S* rRNA gene in the ileum and colon contents of iTfamKO mice by qPCR ($n = 7$ to 9). Data are shown as means \pm SEM, where each dot is a biological sample. P values were determined by unpaired Student's t test. * $P \leq 0.05$.

4. The study completely ignored the fact that cecum is the major site for microbial fermentation in the mice and did not provide any analysis on cecum or caecal contents. In addition, why were SCFAs not quantified from ileal, caecal, and colonic contents in addition to faeces and serum? In particular, what is the effective concentration of butyrate from tributyrin-fed mice in the ileum? This analysis would provide insights into how SCFA composition varies across different intestinal sites in different mouse models, as well as after tributyrin administration.

Following Reviewer 1's suggestions, we now provide a new sets of experiments showing the quantification of SCFAs in ileal, caecal and colonic contents of iTfamKO mice compared to controls. As shown in New Fig. S5A, analyses confirmed a decrease of most SCFAs in the content collected from the cecum and colon of iTfamKO mice, while no gross differences were shown in ileal samples. As pointed by the Reviewer, the concentration of each SCFAs was higher in the cecum than the colon and was marginal in the ileum. These data support a general downregulation of SCFAs across different intestinal compartments of iTfamKO mice and provides further characterization of this mouse model.

Moreover, as suggested by the Reviewer, we have measured the concentration of butyrate in the ileal content of tributyrin (TB)-fed iTfamKO mice in parallel with data presented in Fig. S5A. As shown in **Figure 2 for Reviewers**, the concentration of butyrate in the ileum is significantly increased in iTfamKO mice after TB administration, confirming the efficacy of the treatment in our mouse model.

Figure 2 for Reviewers. Analysis of butyrate in the ileum of iTfamKO mice after tributyrin administration. Quantification of butyrate in samples collected from the ileum of mice ($n = 4$ to 6). Data are shown as means \pm SEM, where each dot is a biological sample. P values were determined by one way analysis of variance (ANOVA). $*P \leq 0.05$.

Minor comments:

1. In Figure 1, please clarify the age of the mice. Are all data panels from 90 days post tamoxifen injection like in 1B?

We do confirm that all the panels in Fig. 1 (with exception of longitudinal studies) and in Fig. S1 are from mice 90-days after tamoxifen administration. We have now clarified this issue in the main text and figure legends.

2. In Figures 1 and 2, the authors showed data for small intestine and from Figure 3 on they begin referring ileum. This creates confusion. Are all the data from ileum? If so, maintaining consistent terminology throughout would help readers follow the results more easily.

We confirm that in Fig. 2, all the data referring to the small intestine were analysed in the ileum (with exception of panel 2D that shows the whole length of the small intestine). We have now unified the terminology throughout the manuscript.

3. In Figures 2L and S3, it is unclear exactly what the heatmap and legend are depicting.

We apologize for the lack of clarity. In these panels, we represent normalized values of the relative expression levels of the genes, as assessed by RT-qPCR, between control and iTfamKO mice (where 100% defines the largest value, and 0% the smaller value in each gene data set). Each column corresponds to a different mouse analysed (e.g. 7 control mice and 9 iTfamKO mice in Fig. 2L). We have now clarified this explanation in the respective figure legends.

4. Fecal output was measured by placing individual mouse in an empty cage for 30 minutes and quantified the number of feces defecated in 30 min. However, this approach might cause stress to the mouse, which can affect gastrointestinal motility. A more reliable approach would be placing individual mouse in metabolic cage and collect feces over a longer duration (overnight), which would provide a more convincing comparison for the defecation output.

As suggested by Reviewer 1, we have now performed overnight individualization of mice to evaluate the defecated output over a longer period of time. As shown in Fig. 2M, the analysis of the number of faeces collected overnight showed that faecal output is notably reduced in iTfamKO mice, suggesting impaired gut motility. We have replaced the previous graph by the one corresponding to the new analysis in the new version of the manuscript and modified materials and methods accordingly (New Fig. 2J).

5. While characterizing intestinal barrier disruption in iTfamKO mice, authors left out the fact that goblet cells and the mucin they produce are crucial for intestinal epithelial barrier function. A protective mucus layer formed by the secreted mucin physically separates the intestinal epithelium from the lumen, preventing the invasion of pathogenic microorganisms. Showing mRNA/protein expression or histological staining for mucins would provide additional evidence of barrier disruption.

We thank Reviewer 1 for raising this interesting point that perfectly fits with our data suggesting gut barrier disruption in iTfamKO mice. Following Reviewer 1's suggestions, we have performed either Periodic Acid-Schiff (PAS) or Alcian Blue staining combined with haematoxylin in ileal and colonic sections from iTfamKO mice to quantify the number and the mucopolysaccharide content of goblet cells. Although the analysis showed that the total number of goblet cells remains unaffected in iTfamKO mice (New Fig. S3), we found an increased mucopolysaccharide content, as assessed by the PAS and Alcian Blue staining, in goblet cells of the ileum and colon of iTfamKO mice compared to controls (New Fig. 2 and S3). Interestingly, we found a decreased frequency of goblet

cells secreting mucus into the ileal lumen, along with a diminished thickness of the mucus layer in the colon of iTfamKO mice compared to control (New Fig.2 and S3).

Altogether, these data suggest that goblet cell function is impaired in iTfamKO mice, resulting in a defective secretion of mucopolysaccharides that may lead to a subsequent decreased mucus layer in iTfamKO mice. These results are in line with recent reports confirming that energy supplied by mitochondrial respiration is required for mucin secretion in the intestine (PMID: 33515804).

6. In Figure 3, the data show that 16s rRNA seq was done on ileum and colon-resident microbiota of control and iTfamKO. In contrast, in Figure 4 microbial analysis of Polg^{mut} mice was carried out using fecal samples. Could the authors clarify the rationale for such sampling differences?

We appreciate Reviewer 1's note. While our laboratory generated a stable colony of iTfamKO mice and their respective controls, we do not have direct access to Polg^{mut} mice. Thanks to a collaboration with the laboratory of Dr. José Antonio Enriquez (Centro Nacional de Investigaciones Cardiovasculares, Madrid, Spain), we obtained previously stored faeces of very old (50-week-old) Polg^{mut} mice and littermate controls, which were precious samples. Although we acknowledge that colonic and faecal microbiota might slightly differ, and even more when samples are obtained in different animal facilities, our aim was to have a general overview of the gut microbiota composition and to clarify if a dysbiotic microbiota was also found in a second mouse model of mitochondrial dysfunction rather than studying whether specific changes in bacterial taxa are common to both mouse models.

7. In Figure 5F, the study shows that FMT restores butyrate levels and to a lesser extent propionate, but has no effect on acetate in the feces of iTfamKO mice. Could the authors explain why only butyrate levels were restored, but not acetate and propionate? Also, in Figure 4F, it is surprising that only butyrate is decreased. The authors should discuss these findings.

As noted by Reviewer 1, faecal acetate levels were not restored by FMT in iTfamKO mice, despite retrieval of butyrate and partial recovery of propionate. A plausible explanation is that butyrate-producing bacteria could use acetate for their growth, diverting acetate into butyrate synthesis. This may paradoxically reduce free acetate pools even if acetogenic bacteria are partially restored by FMT (PMID: 37596118).

As pointed by the Reviewer, in Fig. 4F, we show that only butyrate is significantly decreased in faeces of Polg^{mut} mice. Although there is a slightly partial overlap, different families and genera show specific metabolic pathways for generating different SCFAs. For example, specialized Firmicutes like *Lachnospiraceae* and *Ruminococcaceae* families express the enzymatic machinery necessary to synthesise butyrate. Interestingly, these bacteria are significantly less abundant in faeces of Polg^{mut} mice, correlating with reduced levels of butyrate (Fig. 4G). However, bacteria that normally produce propionate, such as some *Bacteroides spp.* and *Propionibacterium spp.*, and bacteria that are mainly acetogenic, for instance, *Bifidobacterium* or *Lactobacillus spp.*, were either no different or found in a higher abundance in Polg^{mut} mice (Fig. 4G), which could explain the lack of differences in the concentration of these SCFAs. It also needs to be taken into account

that the rate of consumption/absorption by other intestinal bacteria or the host can be altered by the genotype of mice. This has been now included in the Discussion of the main manuscript.

8. The study shows that both control FMT and tributyrin supplementation ameliorate signs of multiorgan morbidity. However, the epigenetic landscape was analyzed only in the context of tributyrin treatment, not following FMT. What was the rationale for this selective approach, given that control microbiota could also potentially restore the altered histone modifications observed in the intestines of iTfamKO mice? Additionally, could the authors clarify whether they consider tributyrin or FMT to be more effective?

Based on our results, we propose that tributyrin treatment is substantially more effective than faecal microbiota transplantation (FMT) into iTfamKO mice. While FMT modestly delays body weight loss, improves muscle strength, and extends maximum lifespan (Fig. 5B, C and G), these beneficial effects are markedly enhanced in tributyrin-treated iTfamKO mice (Fig. 6C, D and I). These differences in efficacy likely reflect distinct mechanisms of action. Tributyrin directly provides sustained source of butyrate, a key microbial-derived metabolite required for host metabolic homeostasis that is deficient in iTfamKO mice, thereby bypassing the need for bacterial colonization and metabolic activity. In contrast, FMT relies on the successful engraftment of a healthy microbiota into an altered intestinal environment, thus limiting its beneficial impact in iTfamKO mice. Regarding this evidence, we propose tributyrin as a more effective intervention than FMT in this mouse model. This has been now included in the Discussion of the main manuscript.

9. The authors should clarify that some articles they cite already link butyrate to increased lifespan (PMID: 28129947) and cite additional articles such as PMID: 25127866.

We have now clarified the link between butyrate and lifespan in different animal models in the Discussion section by citing the suggested articles.

Reviewer #2 (Remarks to the Author):

We kindly thank Reviewer 2 for their time and effort in co-revising our manuscript with Reviewer 1.

Reviewer #3 (Remarks to the Author):

The manuscript “Butyrate Extends Health and Lifespan in Mice with Mitochondrial Deficiency” explores the complex interplay between mitochondrial dysfunction, gut microbiota disruption, and systemic disease. The study is original, well-designed, and carefully executed, with findings that carry significant implications for human health. The manuscript is well written and effectively highlights the broad impact of mitochondrial malfunction on overall physiological homeostasis.

Using two independent models, the authors establish clear associations between mitochondrial failure, microbiota-derived metabolite imbalance, and host health, with a particular focus on gut integrity. Moreover, the study proposes promising therapeutic perspectives for microbial metabolites — notably butyrate and its precursors — as potential clinical tools for patients with mitochondrial disorders.

Overall, this is a valuable and interesting contribution, although several points merit further clarification.

We appreciate Reviewer 3 for their optimistic and constructive comments on our manuscript.

Major Comments

Causality between mitochondrial dysfunction and microbiota disruption

The main conceptual limitation concerns how causality is addressed. At times, the text implies that mitochondrial disease drives microbial dysbiosis. However, detailed examination of the data suggests that the microbiota may itself contribute causally to some of the observed disease outcomes. The therapeutic effects of butyrate further support a primary or parallel contribution of microbiota-driven processes to the phenotype.

Although disentangling these bidirectional interactions is challenging, the data do not convincingly support the notion that microbiota changes are secondary to gut barrier disruption, as proposed. The discussion already hints that microbial alterations may underlie both barrier dysfunction and intestinal pathology; I encourage the authors to expand on this interpretation.

We appreciate Reviewer 3’s comment. The bidirectional relationship between the intestinal barrier and the gut microbiota is indeed complex. Our hypothesis is that metabolic failure due to *Tfam* depletion is the main player in the multimorbidity phenotype of iTfamKO mice, and that secondary alterations in the gut microbiota, through reduced production of SCFAs like butyrate, aggravate but do not cause the phenotype in this mouse model.

To delve into the cause-and-effect between metabolic alterations, gut dysbiosis and the multimorbidity phenotype of iTfamKO mice, we have followed two additional and non-mutually exclusive lines of experiments to characterize how *Tfam* loss affects the intestinal microenvironment.

Firstly, we considered a possibility already discussed in the manuscript, which implies that *Tfam* depletion may impair mitochondrial respiration in intestinal epithelial cells, therefore reducing epithelial oxygen consumption and increasing luminal oxygen availability. Such an increase could remodel microbial communities by disadvantaging obligate anaerobes, including different butyrate producers (Fig. 3). Similar mechanisms have been proposed in other settings (PMID: 27078066, 28798125, 30498100). To test this hypothesis, we assessed intestinal hypoxia using the hypoxyprobe assay, which detects pimonidazole adducts as a readout of tissue hypoxia (see **Figure 3 for Reviewers**). However, we did not detect reduced hypoxia in either the ileum or the colon of iTfamKO mice, suggesting that altered epithelial oxygen consumption is unlikely to be a major driver of dysbiosis in this mouse model.

Figure 3 for Reviewers. Analysis of hypoxia in the intestine of iTfamKO mice. Representative pictures and quantification of hypoxyprobe mean fluorescence in the ileum and the colon of iTfamKO and control mice ($n = 5$ to 6). Nuclei are stained with DAPI. Data are shown as means \pm SEM, where each dot is a biological sample. P values were determined by unpaired Student's t test.

Maintenance of mucus layer homeostasis is another key regulator of gut microbiota. Accordingly, impaired goblet cell function and alterations in the mucus layer have been associated with bacterial dysbiosis in a plethora of mouse models (PMID: 41187059, 26813339, 36425784) and in humans (PMID: 37062177), with special emphasis on the loss of beneficial SCFA-producing bacteria, as we detect in our mouse model (Fig. 3 and new Fig. S5). To explore this second line of research, we analysed the phenotype and function of goblet cells and the mucus layer in iTfamKO mice. In these new set of experiments, we detected a reduced proportion of mucus-secreting goblet cells in the intestinal epithelium and, consistently, a significant shrinking of the intestinal mucus layer in iTfamKO mice compared to controls (New Fig.2 and S3). These data indicate that *Tfam* deficiency in goblet cells impairs mucus secretion, resulting in a defective mucosal barrier. The other way around, the gut microbiota can influence goblet cell differentiation, function and mucus composition (PMID: 40164286, 23692866),

uncovering a possible bidirectional crosstalk. We have now contextualized this in the Discussion section.

Taken together, these data suggest that impaired goblet cell function due to metabolic failure is one of several potential and non-mutually exclusive contributors to gut dysbiosis in iTfamKO mice. Importantly, considering that TFAM deletion occurs in all cells of the host, gut dysbiosis is unlikely to represent the primary cause of multimorbidity in these mice. Our findings support a working model in which gut dysbiosis exacerbates, but does not initiate, the multimorbidity phenotype of iTfamKO mice, since microbiota transplantation from healthy donors or restoration of butyrate levels by TB delays, but does not revert, disease manifestations (Fig. 5 and 6). However, in view of the reported bidirectional crosstalk between the microbiota composition and the integrity of the mucosal layer, a contribution of the dysbiotic microbiome to reported alterations observed in iTfamKO cannot be fully discarded.

Common microbiota signatures

Both models show consistent taxonomic shifts — notably reductions in Lachnospiraceae and Ruminococcaceae in mitochondrial-deficient animals — suggesting a shared “selective pressure” on the microbiota when mitochondrial function is compromised. This point deserves emphasis as it may reveal conserved host–microbiota interactions linked to mitochondrial health.

As Reviewer 3 mentioned, 16S rRNA sequencing analyses of fecal samples in iTfamKO and *Polg^{mut}* mice suggest shared microbiota signatures. *Lachnospiraceae* and *Ruminococcaceae* families are known to be obligate anaerobes that degrade complex carbohydrates through fermentation, resulting in the production of SCFAs like butyrate.

Butyrate-producing bacteria thrive in a healthy mucus layer, consuming mucus-derived components and cross-feeding on metabolites to produce butyrate (PMID: 32917747). Notably, our new data show impaired goblet cell function in the intestine of iTfamKO mice (Fig. S3), which is associated with shifts in the gut microbiota comprising loss of SCFA-producing bacteria. These data align with previous reports suggesting that mitochondrial function is required for the appropriate production of the mucin layer (PMID: 33515804), whose collapse drive the loss of beneficial SCFA-producing bacteria in mice and humans (PMID: 41187059, 26813339, 36425784, 37062177).

Therefore, it is plausible that mitochondrial failure disrupts goblet cell function and mucus production in both iTfamKO and *Polg^{mut}* mice, leading to shared alterations in SCFA-producing bacteria in the intestine. This interesting observation has been now included in the Discussion of the main manuscript.

Additional experimental clarification

Please indicate whether microbiota composition was analyzed at the endpoint of the FMT experiments. How does mitochondrial deficiency influence microbial profiles when control feces are transplanted into Tfam-KO mice? To further delineate causality, a reciprocal FMT experiment transferring Tfam-KO microbiota into WT mice could provide valuable insight. Although the current design is elegant and convincing, the

strong mitochondrial phenotype may mask microbiota-derived contributions to disease outcomes.

In light of the significantly better effects exerted by the TB treatment in comparison with FMT strategies, we agreed with the editor to preferentially focus our efforts in trying to characterize the first strategy (e.g. Fig 6). Nevertheless, to address the reviewer question on the potential role of the gut microbiota in the phenotype of iTfamKO mice, we have performed a new set of reciprocal FMT experiments suggesting that transplantation of microbiota from donor iTfamKO mice into recipient healthy control mice is not sufficient to induce noticeable signs of multimorbidity in these animals. In brief, we transplanted microbiota from either control or iTfamKO mice into control mice that were subsequently administered with dextran sulfate sodium (DSS) to evaluate the pathogenic potential of a transplanted microbiota (PMID: 34818535, 31107248, 38553486). The analysis did not reveal any differences in terms of colon length, percentage of body weight loss, rectal bleeding score, or disease activity index regardless of the transplanted microbiota (see **Figure 4 for Reviewers**). We neither observed any evident signs of multimorbidity in control mice undergoing FMT from iTfamKO mice before DSS treatment.

Figure 4 for Reviewers. Fecal microbiota from iTfamKO mice does not worsen DSS-induced colitis. (A) Experimental design of DSS administration to control mice transplanted with microbiota from either control (FMT_{Control}) or iTfamKO (FMT_{KO}) mice. (B) Longitudinal assessment of relative body weight following DSS administration ($n = 4$ to 8). (C) Longitudinal assessment of rectal bleeding score (negative=0, visual pellet bleeding=2; gross bleeding or around anus=4) following DSS administration ($n = 4$ to 8). (D) Longitudinal assessment of disease activity index taking into account body weight loss (no loss=0, < 5%= 1; 5-10%= 2; 10-20%= 3; > 20%= 4), stool consistency (normal=0; loose=2; diarrhoea=4), and rectal bleeding score following DSS administration ($n = 4$ to 8). (E) Representative picture and quantification of colon length ($n = 4$ to 8). Data are shown as means \pm SEM, where each dot is a biological sample. P values were determined by (B, C and D) two-way analysis of variance (ANOVA) and (E) one-way ANOVA with Tukey's multiple comparison test. ** $P \leq 0.01$; *** $P \leq 0.001$; **** $P \leq 0.0001$.

These results suggest that (1) the phenotype of iTfamKO mice primarily stems from mitochondrial dysfunction resulting from the loss of TFAM, and (2) dysbiotic microbiota

communities in iTfamKO mice are not intrinsically harmful, at least in a DSS-induced colitis paradigm. We propose that the loss of microbiota-derived metabolites essential for the host, such as butyrate, aggravates signs of multimorbidity, especially in the context of mitochondrial deficiency and not upon mitochondrial sufficiency. This is supported by previous observations showing higher reliance on microbiota derived by-products in models of mitochondria-related neurodegeneration (PMID: 31330533) and models of progeria (PMID: 31332389). Accordingly, transplantation of control microbiota, which restores butyrate production, or direct administration of butyrate precursors delays the multimorbidity phenotype in iTfamKO mice.

Co-housing experiments could offer a more physiological means to test transmissibility, avoiding potential confounding from antibiotic pretreatment. Even if not performed, please comment on whether such an approach was considered.

We thank Reviewer 3's comment. We acknowledge that both co-housing and FMT strategies have been found efficacious in terms of microbiota transmissibility. Given the strong phenotype shown by iTfamKO mice, we chose FMT (1) to avoid unsolicited transfer of microbiota from iTfamKO mice to control mice by coprophagia, in other words, to have uncontaminated control donor mice, and (2) to control the quantity and frequency of microbiota transfer during the experiment, both of which cannot be regulated just by co-housing mice.

Regarding potential confounding effects of antibiotics, we believe that short antibiotic administration (three days) is unlikely to have had a beneficial outcome. Moreover, we performed a new set of experiments using the same cocktail of antibiotics for a longer period of time (from day 30 post-tamoxifen onwards). We found that this protocol significantly shortened the lifespan of iTfamKO mice (see **Figure 5 for Reviewers**), discarding a potentially beneficial effect of antibiotics. This suggests that iTfamKO mice are especially sensitive to microbiota depletion as previously observed in mouse models of neurodegeneration induced by a mitochondrial dysfunction (PMID: 31330533).

Figure 5 for Reviewers. Long-term antibiotic treatment shortens iTfamKO mouse lifespan. (A) Experimental design of antibiotic (Abx) or vehicle (Veh) administration to control and iTfamKO mice. (B) qPCR quantification of total bacterial DNA in faeces ($n = 5$ to 6). (C) Heatmap depicting normalized values of short-chain fatty acid (SCFA) levels in faeces ($n = 5$ to 6). (D) Kaplan-Meier survival curves ($n = 6$ to 14). Data are shown as means \pm SEM, where each dot is a biological sample. P values were determined by (B and C) unpaired Student's t test and (D) log-rank (Mantel-Cox) test. * $P \leq 0.05$; ** $P \leq 0.01$ and *** $P \leq 0.001$.

Minor Comments

In Tfam-KO mice, several SCFA reductions (acetate and propionate in serum; isovalerate in feces) are not statistically significant, yet the text does not distinguish between significant and non-significant changes. Please clarify.

We thank Reviewer 3's comment. We have now clarified that statement in the Results section of the manuscript.

Previous work has linked mitochondrial dysfunction to microbiota alterations and secondary SCFA shifts (e.g., increased mitochondrial activity in MCJ-KO mutants). It would strengthen the discussion to acknowledge some of these studies:

Houghton D, et al. *J Gerontol A Biol Sci Med Sci*. 2018; 73(5):571-578.
Juárez-Fernández M, et al. *Hepatology*. 2023; 77(5):1654-1669.
Peña-Cearra A, et al. *Gut Microbes*. 2023; 15(2):2266626.

Following Reviewer 3's suggestion, we have now cited these studies in the Discussion section of the manuscript.